# A unifying mechanism governing inter-brain neural relationship during social interactions

**Wujie Zhang\*, Maimon C Rose, Michael M Yartsev\***

Helen Wills Neuroscience Institute and Department of Bioengineering, University of California, Berkeley, Berkeley, United States

**Abstract** A key goal of social neuroscience is to understand the inter-brain neural relationship—the relationship between the neural activity of socially interacting individuals. Decades of research investigating this relationship have focused on the similarity in neural activity across brains. Here, we instead asked how neural activity differs between brains, and how that difference evolves alongside activity patterns shared between brains. Applying this framework to bats engaged in spontaneous social interactions revealed two complementary phenomena characterizing the inter-brain neural relationship: fast fluctuations of activity difference across brains unfolding in parallel with slow activity covariation across brains. A model reproduced these observations and generated multiple predictions that we confirmed using experimental data involving pairs of bats and a larger social group of bats. The model suggests that a simple computational mechanism involving positive and negative feedback could explain diverse experimental observations regarding the inter-brain neural relationship.

**\*For correspondence:** wujie@berkeley.edu (WZ); myartsev@berkeley.edu (MMY)

**Competing interest:** The authors declare that no competing interests exist.

## Introduction

What is the relationship between the neural activity of socially interacting individuals? This central question in social neuroscience has motivated nearly two decades of research spanning a diversity of species and methodologies (e.g. *Babiloni and Astolfi, 2014*; *Dumas et al., 2011*; *Freiwald, 2020*; *Hasson et al., 2012*; *Hasson and Frith, 2016*; *Hoffmann et al., 2019*; *Kingsbury and Hong, 2020*; *Koike et al., 2015*; *Konvalinka and Roepstorff, 2012*; *Liu et al., 2018*; *Montague et al., 2002*; *Redcay and Schilbach, 2019*; *Scholkmann et al., 2013*; *Schoot et al., 2016*; *Testard et al., 2021*; *Tseng et al., 2018*; *Wass et al., 2020*). Yet, despite this diversity, nearly all research has tackled the study of inter-brain relationship from a single perspective: considering the neural activity of two interacting individuals as the two variables of interest and searching for similarities between them. Commonly, similarity was assessed using measures related to either correlation (*Dikker et al., 2014*; *Kawasaki et al., 2013*; *King-Casas et al., 2005*; *Kingsbury et al., 2019*; *Kinreich et al., 2017*; *Levy et al., 2017*; *Liu et al., 2017*; *Montague et al., 2002*; *Piazza et al., 2020*; *Silbert et al., 2014*; *Spiegelhalder et al., 2014*; *Stephens et al., 2010*; *Stolk et al., 2014*; *Tomlin et al., 2006*; *Zadbood et al., 2017*; *Zhang and Yartsev, 2019*) or coherence (*Cui et al., 2012*; *Dikker et al., 2017*; *Dumas et al., 2010*; *Goldstein et al., 2018*; *Levy et al., 2017*; *Lindenberger et al., 2009*; *Montague et al., 2002*; *Mu et al., 2017*; *Stolk et al., 2014*; *Yang et al., 2020*; *Yun et al., 2012*).

However, one aspect of the inter-brain neural relationship has remained unexplored: the difference in neural activity between brains. We reasoned that inter-brain difference is more than simply a lack of inter-brain correlation. Specifically, the detailed dynamics of the difference between brains could contain information not captured by measures of inter-brain similarity such as correlation. Therefore, here we took an approach that focused on both the similarity and difference in inter-brain activity, and on how the two co-evolve over time. We applied this approach to neural activity simultaneously

recorded from socially interacting Egyptian fruit bats (*Rousettus aegyptiacus*), a mammalian species known for its high level of sociality (*Herzig-Straschil and Robinson, 1978*; *Harten et al., 2018*; *Prat et al., 2015*; *Prat et al., 2016*; *Prat et al., 2017*; *Omer et al., 2018*; *Cvikel et al., 2015*; *Egert-Berg et al., 2018*; *Kwiecinski and Griffiths, 1999*). Our recordings targeted the frontal cortex (*Figure 1— figure supplement 1*), a region implicated in social cognition in rodents, bats, non-human primates, and humans (*Adolphs, 2001*; *Amodio and Frith, 2006*; *Cao et al., 2018*; *Chang et al., 2013*; *Eliades and Miller, 2017*; *Forbes and Grafman, 2010*; *Haroush and Williams, 2015*; *Kingsbury et al., 2019*; *Liang et al., 2018*; *Miller et al., 2015*; *Nummela et al., 2017*; *Ong et al., 2021*; *Pearson et al., 2014*; *Rose et al., 2021*; *Rudebeck et al., 2008*; *Tremblay et al., 2017*; *Zhang and Yartsev, 2019*; *Zhou et al., 2017*). We then used modeling to better understand the experimentally observed inter-brain relationship, which in turn produced a number of predictions that we subsequently confirmed using experimental data. Combined, the insights from the experimental and modeling approaches provide a unifying explanation to a range of experimental observations regarding the inter-brain relationship during social interactions.

## Results

### Inter-brain relationships in social and non-social contexts

We performed wireless extracellular neural recording simultaneously from pairs of bats. We recorded four types of neural signals: local field potential (LFP) power in the 30–150 Hz and 1–29 Hz bands (previously identified as relevant frequency bands in bat frontal cortical LFP [*Zhang and Yartsev, 2019*]), multiunit activity, and single unit activity (Materials and methods). The analyses in the main text focus on 30–150 Hz LFP power, as previous work has shown that this frequency range exhibits strong inter-brain correlation in bats (*Zhang and Yartsev, 2019*).

To compare the inter-brain relationship between social and non-social conditions, neural activity was recorded in two sets of experiments. In one experiment, pairs of bats behaved freely and interacted with each other inside a chamber ('one-chamber sessions'; *Figure 1A*). In the second set of experiments, the same bats freely behaved in separate, identical chambers ('two-chambers sessions'; *Figure 1B*). There were three types of two-chambers sessions: (1) two bats each freely behaving in isolation; (2) two bats each freely behaving in the presence of identical auditory stimuli (playback of bat calls); (3) two bats each freely behaving and interacting with a different partner in separate chambers.

Plotting the neural activity of the two bats in one-chamber sessions shows a high degree of similarity between brains (*Figure 1C*), which is not the case for two-chambers sessions (*Figure 1D*), as demonstrated previously (*Zhang and Yartsev, 2019*). Such plots highlight the degree of inter-brain similarity, but makes it easy to overlook the detailed dynamics of the difference between brains. We therefore sought an analysis framework that would enable us to explicitly examine inter-brain difference and similarity side by side.

### Relative magnitudes and timescales of inter-brain difference and mean components

When studying the inter-brain relationship, two typical variables of interest are the neural activity of each brain (e.g. in *Figure 1C*, each variable is the normalized LFP power averaged across the recording channels of one brain). We can represent them as a two-dimensional vector $\begin{pmatrix} a_1(t) \\ a_2(t) \end{pmatrix}$, where $a_1(t)$ and $a_2(t)$ are the activity of bat 1 and bat 2 at time $t$, respectively. Through a change of basis, the same activity can be represented under another orthogonal basis as the mean and difference between the two brains: $\begin{pmatrix} a_M(t) \\ a_D(t) \end{pmatrix} = \frac{1}{2} \begin{pmatrix} a_1(t) + a_2(t) \\ a_1(t) - a_2(t) \end{pmatrix}$, where $a_M(t)$ is the mean component of the activity, and $a_D(t)$ is the difference component (here the difference component is defined as $\frac{1}{2}\left[a_1(t) - a_2(t)\right]$ rather than $a_1(t) - a_2(t)$ so as to have the same scale as the mean component). $a_M(t)$ represents the common activity pattern shared between the brains, that is, what is similar between the brains; $a_D(t)$ represents the moment-to-moment activity difference between the brains.

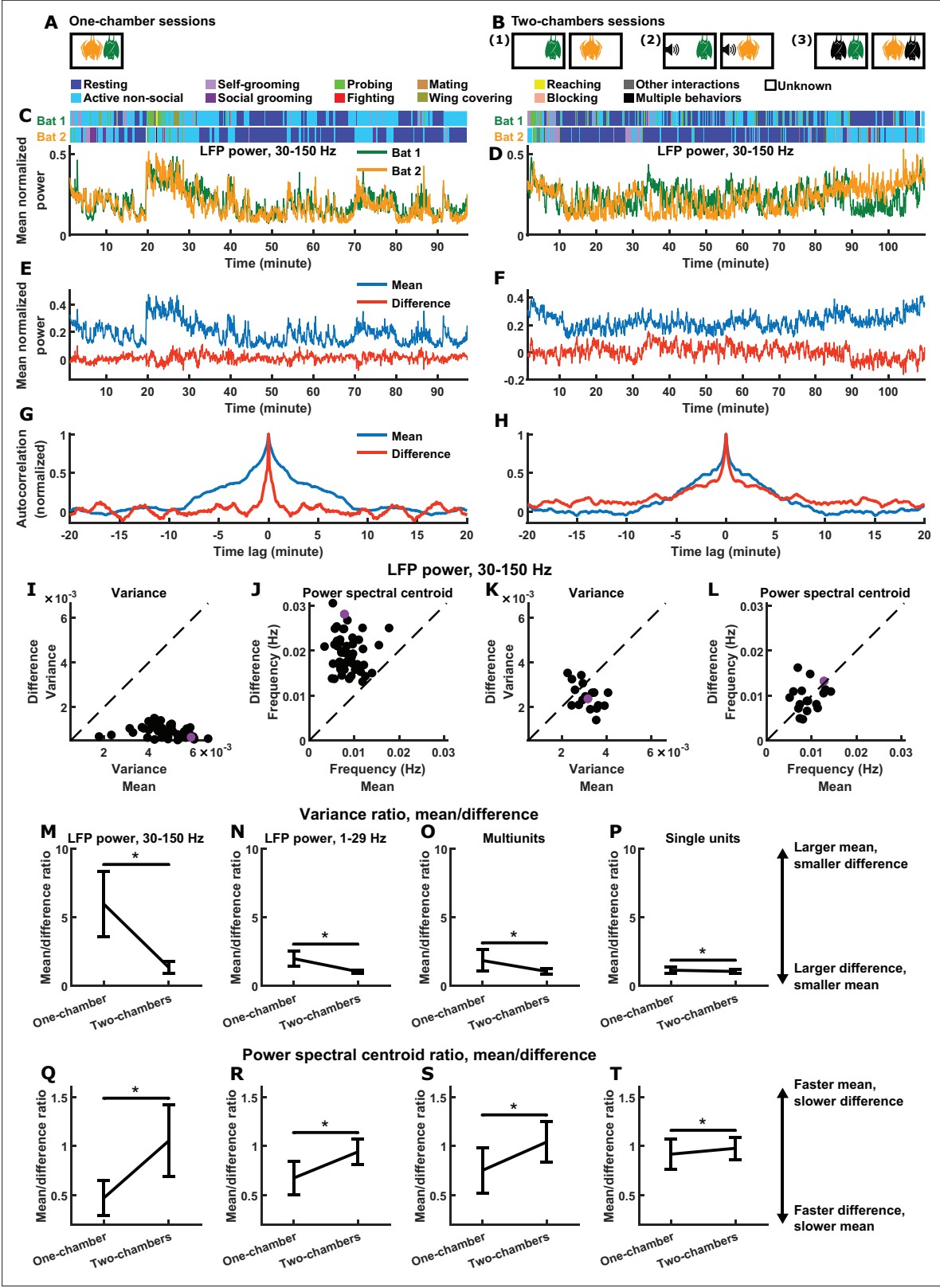

**Figure 1.** Relative magnitudes and timescales of the inter-brain difference and mean components. (**A**) In each one-chamber session, two bats freely interacted with each other while neural activity was wirelessly recorded simultaneously from their frontal cortices. (**B**) In each two-chambers session, the same bats from the one-chamber sessions freely behaved in separate, identical chambers, while neural activity continued to be simultaneously and wirelessly recorded from their frontal cortices. Two-chambers sessions included three conditions: (1) two bats each freely behaving in isolation; (2)

*Figure 1 continued*

two bats each freely behaving while listening to identical auditory stimuli; (3) two bats each freely behaving and interacting with a different partner in separate chambers. (**C**)-(**D**) Mean normalized LFP power in the 30–150 Hz band (Materials and methods), averaged across all channels for each bat, on an example one-chamber session (**C**) and an example two-chambers session (**D**). Shown above are the behaviors of the two bats as a function of time, which were manually annotated frame-by-frame from recorded video. The example two-chambers session was of the third type illustrated in (**B**). (**E**) The neural activity of the two bats from (**C**) after a change of basis, showing the mean and difference between bats. At a given time *t*, the mean and difference components are defined as $\frac{1}{2}\left[\boldsymbol{a}_1\left(t\right) + \boldsymbol{a}_2\left(t\right)\right]$ and $\frac{1}{2}\left[\boldsymbol{a}_1\left(t\right) - \boldsymbol{a}_2\left(t\right)\right]$, respectively, where $\boldsymbol{a}_1\left(t\right)$ and $\boldsymbol{a}_2\left(t\right)$ are respectively the neural activity of bat 1 and bat 2 plotted in (**C**). Note that the mean component had a large variance, whereas the difference component had a small variance, hovering around zero. (**F**) Same as (**E**), but for the example session from (**D**). Note that the variances are more comparable between the mean and difference components. (**G**) Autocorrelations (peak-normalized) of the mean and difference components shown in (**E**). The autocorrelations were computed after subtracting from each time series its average over time. Note that the difference component varied on faster timescales than the mean component. (**H**) Same as (**G**), but for the example session from (**D**). Note that the timescales are more comparable between the mean and difference components. (**I**)-(**J**) Variance (**I**) and power spectral centroid (**J**) of mean normalized 30–150 Hz LFP power, for the mean and difference components. Each dot is a single one-chamber session (the purple dot is the session shown in (**C**), (**E**), and (**G**)). Variance quantifies activity magnitude, and power spectral centroid quantifies timescale (higher centroids mean faster timescales). Note that, on every one-chamber session, the difference component was smaller and faster than the mean component. The dotted lines are unity. Note that the power spectral centroid was calculated from time series of mean normalized LFP power (e.g., as plotted in (**E**)), not from time series of LFP itself. (**K**)-(**L**) Same as (**I**)-(**J**), but for two-chambers sessions. The purple dot is the session shown in (**D**), (**F**), and (**H**). The mean and difference components have comparable magnitudes and timescales in the two-chambers sessions. (**M**)-(**P**) The average variance ratio (mean component variance divided by difference component variance) for mean normalized 30–150 Hz LFP power (**M**), mean normalized 1–29 Hz LFP power (**N**), multiunits (**O**), and single units (**P**). The averages were taken across sessions for LFP power, and across unit pairs (pooled from all sessions) for multiunits and single units. Error bars denote standard deviations. *, p < 0.05, Wilcoxon rank sum test. (**Q**)-(**T**) Same as (**M**)-(**P**), but for average power spectral centroid ratio (mean component centroid divided by difference component centroid). Note that, for all four neural signals, the difference component was smaller and faster than the mean component on one-chamber sessions. See *Figure 1—figure supplements 2–4* for examples and detailed results for 1–29 Hz LFP power, multiunits, and single units.

The online version of this article includes the following figure supplement(s) for figure 1:

**Figure supplement 1.** Neural recording location.

**Figure supplement 2.** Inter-brain difference and mean components: 1–29 Hz LFP power.

**Figure supplement 3.** Inter-brain difference and mean components: multiunit activity.

**Figure supplement 4.** Inter-brain difference and mean components: single unit activity.

---

Note that the two components can vary independently—a large (or small) $\mathbf{a}_M\left(t\right)$ does not necessarily imply a small (or large) $\mathbf{a}_D\left(t\right)$ .

*Figure 1E* shows the mean and difference activity components for the example session from *Figure 1C*. Visualizing the two components side-by-side, two relationships are immediately apparent. First, the mean component had much larger variance than the difference component. This is expected given the high degree of similarity in neural activity between the brains: while the variances are not entirely determined by inter-brain correlation, a positive correlation does mathematically imply larger variance for the mean compared to the difference component (see Materials and methods for details). Second, the difference component evolved over time at much faster timescales than the mean component. This is even more apparent when examining the autocorrelations of the mean and difference components (*Figure 1G*), where the narrower autocorrelation of the difference indicates that it varied faster than the mean.

In contrast, the two-chambers sessions showed a very different picture. *Figure 1F* shows the mean and difference components for the example session from *Figure 1D*. Here, the variances of the two components were comparable, and so were their timescales, which is also apparent from their comparable autocorrelations shown in *Figure 1H*.

We next quantified the magnitudes and timescales of the mean and difference components for 30–150 Hz LFP power on all sessions, using variance as a measure for magnitude, and power spectral centroid as a measure for timescale (*Figure 1I–L*). The power spectral centroid was calculated as follows. Given the time series of an activity component, we subtracted from it its average over time before calculating its power spectrum, and then computed the weighted average of the frequencies, where each frequency was weighted by the power at that frequency (Materials and methods). The relationships seen in the one-chamber example above was robust: on every one-chamber session, the difference component had smaller magnitudes (*Figure 1I*) and faster timescales (*Figure 1J*) than the mean component (note that the relative magnitudes are a reflection of inter-brain correlation). On

two-chambers sessions, in contrast, the mean and difference were comparable (*Figure 1K–L*). These relationships are summarized in *Figure 1M–T* for all four neural signals: LFP power in the 30–150 Hz and 1–29 Hz bands, multiunits, and single units all showed the same significant trends (see *Figure 1—figure supplements 2–4* for examples and detailed results for 1–29 Hz LFP power, multiunits, and single units).

Thus, the inter-brain neural relationship during social interactions is characterized by two robust signatures of the mean and difference components: their relative magnitudes (which reflects inter-brain correlation) and their relative timescales. As mentioned above, the observed relative magnitudes are mathematically implied by the presence of inter-brain correlation (Materials and methods). It is important to determine whether the observed relative timescales are also implied by inter-brain correlation and the observed relative magnitudes. As we show in Materials and methods (section 'Surrogate data and the relationship between inter-brain correlation and mean and difference components'), the answer is no: having a given combination of correlation, mean component variance, and difference component variance does not place constraints on the timescales of the two components. Specifically, it does not constrain the difference to be faster than the mean. To explicitly demonstrate this, we constructed a set of surrogate data as a counter-example (*Figure 2*). The surrogate data was designed to have inter-brain correlation, mean component variance, and difference component variance that are identical to the actual data. However, unlike the actual data, the surrogate data was designed to have difference components that were slower than the mean components, the opposite of what was experimentally observed (*Figure 2C and E*). Thus, the relative timescales of the difference and mean components are not dictated by their relative magnitudes or by inter-brain correlation.

What, then, might explain the robust relationships observed between the timescales and magnitudes? And are there separate mechanisms responsible for the observed relative timescales on the one hand, and the observed relative magnitudes (and the related phenomenon of inter-brain correlation) on the other? To address these questions, we next model the observed neural activity to infer the computational mechanisms governing the inter-brain difference and mean components, and by extension, mechanisms underlying inter-brain correlation.

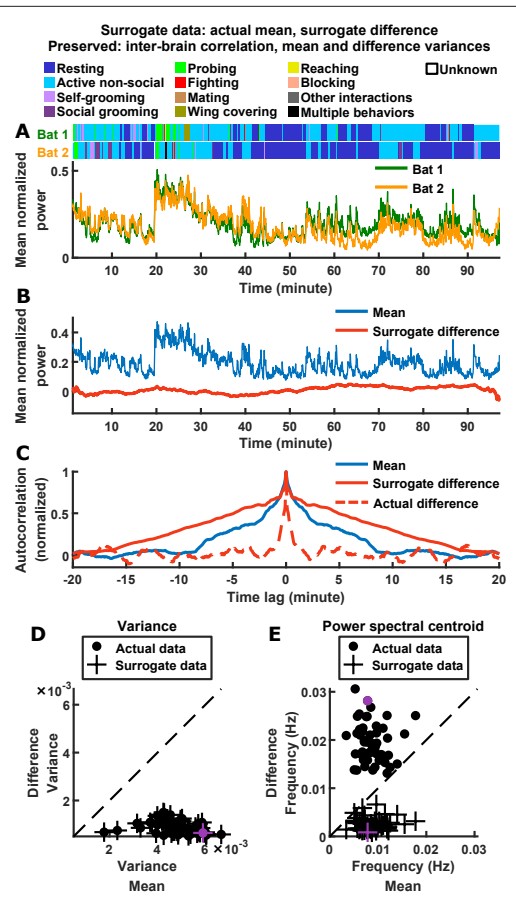

**Figure 2.** Relative timescales of the difference and mean components are not determined by their relative magnitudes or levels of inter-brain correlation. (**A**) Surrogate data generated from the actual data shown in *Figure 1C*, by combining the actual mean component with a surrogate difference component (Materials and methods). The surrogate was tailored such that the difference component variance and inter-brain correlation of the original experimental data were preserved (the mean component variance was also preserved since the actual mean component was used in the surrogate data). Shown above are the actual behaviors of the two bats replotted from *Figure 1C*. (**B**) The surrogate data from (**A**) plotted as the activity of its mean and difference components. The mean here is identical to the actual mean in *Figure 1E*, while the difference here has the same variance as the actual difference in *Figure 1E*, but with slower timescales. (**C**) The autocorrelations (peak-normalized) of the mean and difference components shown in (**B**) and of the actual difference component shown in *Figure 1E*. The autocorrelations were computed after subtracting from each time series its average over time. Note that for the surrogate data, the difference was slower than the mean, the opposite of what was observed experimentally. (**D**)-(**E**) Variance (**D**) and power spectral centroid (**E**) of mean normalized 30–150 Hz LFP power, for the mean and difference components of the actual

*Figure 2 continued on next page*

*Figure 2 continued*

data and surrogate data. Each dot is actual data from a single one-chamber session (replotted from *Figure 1I–J*), and each plus is surrogate data generated from the actual data of a single one-chamber session (purple dots and pluses denote the example session shown in *Figure 1C* and (**A**)-(**C**)). The dotted lines are unity. The surrogate data preserve the actual inter-brain correlations (Materials and methods), as well as the variances of the actual mean and difference components (pluses in (**D**) are at the same positions as the dots), but have slower difference components than mean components (pluses below the unity line in (**E**)).

## Model suggests a feedback mechanism governing inter-brain neural relationship

We modeled the neural activity of two bats using a linear differential equation (see *Figure 3A* for the equation; *Dayan and Abbott, 2005*). In the model, the neural activity variables interact through the functional coupling matrix $C = \begin{pmatrix} -C_S & C_I \\ C_I & -C_S \end{pmatrix}$, where $C_S$ is the strength of functional self-coupling and $C_I$ is the strength of functional across-brain coupling. When modeling one-chamber sessions, $C_I > 0$, so that the activity of each bat influences the other bat's activity. This functional across-brain coupling models the effects of sensorimotor interactions and attentional processes (*Hasson et al., 2012*): for example, when bat 1's neural activity increases due to its active movements (*Gervasoni et al., 2004*; *McGinley et al., 2015*), the movements create sensory inputs to bat 2, which can drive bat 2's neural activity to the extent that bat 2 is paying attention (*Chun et al., 2011*; *Driver, 2001*; *Fritz et al., 2007*; *Reynolds and Chelazzi, 2004*). On the other hand, when modeling two-chambers sessions, $C_I = 0$, so that the activity of each bat does not influence the other bat's activity. Furthermore, in both one-chamber and two-chambers models, the activity of each bat is modulated by its own behavior, which is simulated using Markov chains based on empirical behavioral transition frequencies (Materials and methods). Note that the usage of the term 'functional across-brain coupling' in our model should be distinguished from the sense it is sometimes used in the literature to simply denote a presence of neural correlation or coherence across brains (e.g. *Levy et al., 2017*).

This model was used to simulate neural activity, which was then analyzed using the same methods used to analyze experimental data. The model qualitatively reproduces the experimentally observed neural activity patterns: simulated activity shows variability over time, modulation by behavior, and correlation across brains (*Figure 3B*; see *Figure 3—figure supplement 1* for a quantitative comparison between model and data). More importantly, the model also qualitatively reproduces all the experimentally observed relationships between the magnitudes and timescales of the difference and mean components, as can be seen in the activity and autocorrelation of the two components on an example simulation (*Figure 3C–D*). Specifically, the difference component had smaller magnitudes and faster timescales than the mean component in the one-chamber model, but not in the two-chambers model (*Figure 3C–F*).

What mechanisms in the model are responsible for these results? We now analyze the model to answer this (see Materials and methods for the detailed analysis). The model describes the evolving neural activity of two bats and uses separate variables to represent the activity of each bat. This is a basis in which the neural activity variables are coupled to each other in the one-chamber model: the activity of bat 1 influences the activity of bat 2, which in turn feeds back on the activity of bat 1. It is much easier to understand how activity evolves if we change to a basis in which the activity variables are uncoupled. This basis consists of the eigenvectors of the functional coupling matrix $C$. Under the eigenvector basis, the neural activity variables become the mean activity across bats, and the difference in activity between bats (*Figure 3G*). Thus, the uncoupled activity variables are precisely our variables of interest: the inter-brain mean and difference components. Each of the two components provides feedback onto itself, with feedback strengths being the eigenvalues of $C$: $-(C_S - C_I)$ for the mean component and $-(C_S + C_I)$ for the difference component. In the two-chambers model, $C_I = 0$, so the two components receive equal feedback. In the one-chamber model, $C_I > 0$, so functional across-brain coupling acts as positive feedback for the mean component, which amplifies the mean component while slowing it down. On the other hand, functional across-brain coupling acts as negative feedback for the difference component, which suppresses the difference component while speeding it up (*Figure 3H*). Thus, in the model, a single mechanism—opposite feedback, that is,

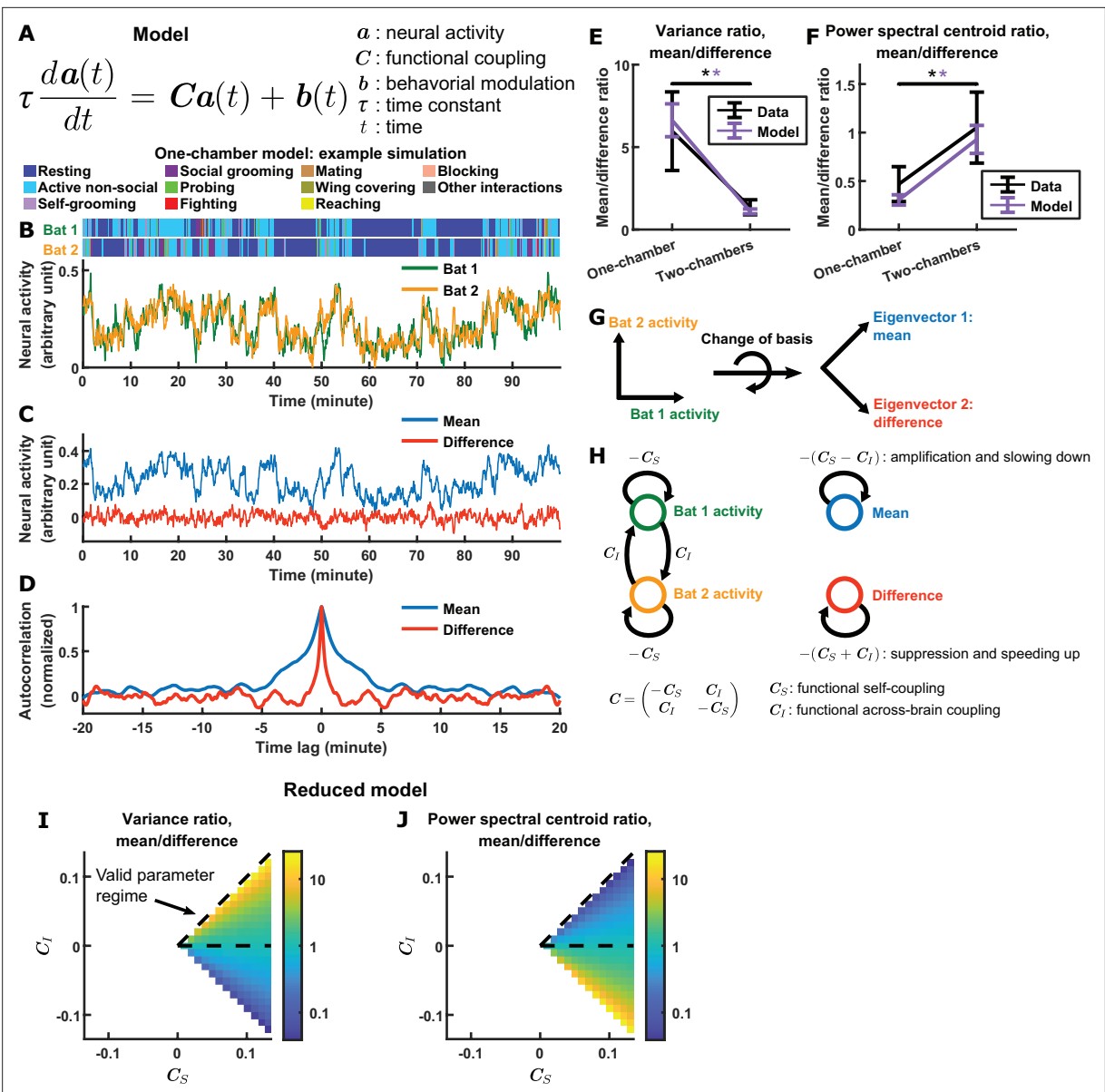

**Figure 3.** Model explains relationship between difference and mean. (**A**) The evolving neural activity of two bats are modeled by a linear differential equation. $\boldsymbol{a}(t) = \begin{pmatrix} a_1(t) \\ a_2(t) \end{pmatrix}$ is the activity of bat 1 and bat 2 at time $t$. $\boldsymbol{C} = \begin{pmatrix} -C_S & C_I \\ C_I & -C_S \end{pmatrix}$ is the functional coupling matrix, where $C_S$ is the strength of functional self-coupling and $C_I$ is the strength of functional across-brain coupling (note that functional across-brain coupling obviously should not be interpreted as direct coupling via actual neural connections). $\boldsymbol{b}(t) = \begin{pmatrix} b_1(t) \\ b_2(t) \end{pmatrix}$ is the modulation of each bat's activity by its behaviors, where the behaviors are simulated using Markov chains. See Materials and methods for details on the model and for the values of the parameters. (**B**) Simulated neural activity and behaviors from an example one-chamber simulation. (**C**) The simulated activity from (**B**) plotted as the activity of its mean and difference components. Note the smaller magnitude of the difference compared to the mean. (**D**) The autocorrelations (peak-normalized) of the mean and difference components shown in (**C**). The autocorrelations were computed after subtracting from each time series its average over time. Note that the difference varied on faster timescales than the mean. (**E**)-(**F**) The average variance ratio (**E**; mean component variance divided by difference component variance) and average power spectral centroid ratio (**F**; mean component centroid divided by difference component centroid), for model simulations (purple) or mean normalized 30–150 Hz LFP power from the data (black). The averages were taken across simulations for the model, and across sessions for the data. Error bars denote standard deviations. *, p < 0.05, Wilcoxon rank sum test. (**G**)-(**H**) Model mechanism. See Materials and methods for details. (**G**) Schematic illustrating a change of basis for representing the model. The model describes the neural activity of two bats evolving in a 2D space. Activity in this space can be represented using any choice of axes. The equation in (**A**) assumes the axes illustrated on the left,

*Figure 3 continued on next page*

*Figure 3 continued*

where each axis is the neural activity of one bat. After a rotation of 45 degrees, the axes now represent the mean and difference components (illustrated on the right), which are also the eigenvectors of the functional coupling matrix $\boldsymbol{C}$. (**H**) The neural activity variables of the two bats are coupled to each other (left). Changing to the eigenvector basis transforms them into uncoupled variables: the mean and difference components (right). The mean and difference components each provides feedback onto itself, with feedback strengths being the eigenvalues of $\boldsymbol{C}$, which depend on the functional across-brain coupling $\boldsymbol{C_I}$. In the one-chamber model, $\boldsymbol{C_I} > 0$, so functional across-brain coupling amplifies and slows down the mean component through positive feedback, and suppresses and speeds up the difference component through negative feedback. (**I**)-(**J**) The dependence of variance ratio (**I**) and power spectral centroid ratio (**J**) on the coupling strength parameters, for a reduced model where the behavioral modulation for the mean and difference components have identical, flat power spectra. The variance ratio and centroid ratio are approximately $\frac{C_S+C_I}{C_S-C_I}$ and $\frac{C_S-C_I}{C_S+C_I}$, respectively; thus, the parameter regime that qualitatively reproduces experimental observations is the regime of positive functional across-brain coupling $\boldsymbol{C_I}$ (delimited by dashed lines). The white spaces are regions of the parameter space where the model is unstable. Note that the color maps are in log scale. See Materials and methods for details.

The online version of this article includes the following figure supplement(s) for figure 3:

**Figure supplement 1.** Comparing actual and simulated neural activity as a function of time.

**Figure supplement 2.** Behavior models.

positive feedback to the mean component and negative feedback to the difference component—contributes to all of our observations relating the magnitudes and timescales of the mean and difference components.

We can gain a more precise understanding of the dependence of model output on functional coupling, by analyzing a reduced version of the model. The reduced model assumes a simplified structure for behavioral modulation, which allows derivation of simple analytic expressions for the variance ratio and power spectral centroid ratio of the mean and difference components (Materials and methods). Specifically, the variance ratio and centroid ratio can be shown to be approximately $\frac{C_S+C_I}{C_S-C_I}$ and $\frac{C_S-C_I}{C_S+C_I}$, respectively (*Figure 3I–J*). This simple dependence on the functional coupling parameters $C_S$ and $C_I$ is consistent with our analysis of the full model above: functional across-brain coupling $C_I$ modulates the mean and difference components in opposite directions. Furthermore, it identifies the parameter regime in which the experimental observations lie—the parameter regime of positive $C_I$ (*Figure 3I–J*).

In the full version of our model, in addition to functional coupling, behavioral modulation also contributes to the inter-brain relationship (the effects of behavioral modulation are analyzed in depth in Materials and methods and *Figure 3—figure supplement 2*). In particular, the tendency of bats to engage in the same behavior at the same time (*Figure 3—figure supplement 2G*) contributes to inter-brain correlation. This led us to ask whether this form of behavioral coordination alone is sufficient to explain experimental data in the absence of functional across-brain coupling. One specific experimental observation suggests that the answer is no: *Figure 4A, B and G* show that, after removing time periods of coordinated behavior from experimental sessions, inter-brain correlation persisted. We found that our model can reproduce this phenomenon (*Figure 4C, D and G*). Next, we simulated an alternative model with behavioral coordination, but without functional across-brain coupling (Materials and methods). *Figure 4E–G* show that, in the absence of functional across-brain coupling, inter-brain correlation disappeared after removing time periods of coordinated behaviors. Thus, in the framework of our model, functional across-brain coupling is necessary to reproduce the experimental results.

In our model, functional across-brain coupling took a simple form: the activity variables of the two brains are linearly coupled. We made this modeling choice for its simplicity, but one may wonder, does the form of functional across-brain coupling matter, or is any form of coupling capable of reproducing the experimental results? While we cannot exhaustively test all alternative ways of modeling functional coupling, we sought to explore one alternative, the Kuramoto model, because it is a well-known model of coupling between oscillatory variables (*Strogatz, 2000*; *Acebrón et al., 2005*). The Kuramoto model has been widely used to model diverse synchronization phenomena in physics, chemistry, and biology, including neural synchronization both within and across brains (e.g. *Breakspear et al., 2010*; *Cumin and Unsworth, 2007*; *Dumas et al., 2012*; *Schmidt et al., 2015*). Here, we adapted it to model inter-brain synchronization in bats. In this model, the fluctuating neural activity of the interacting bats are abstracted as oscillators whose phases are dynamically coupled depending on their phase difference (Materials and methods). This phase-coupling mechanism is able to reproduce

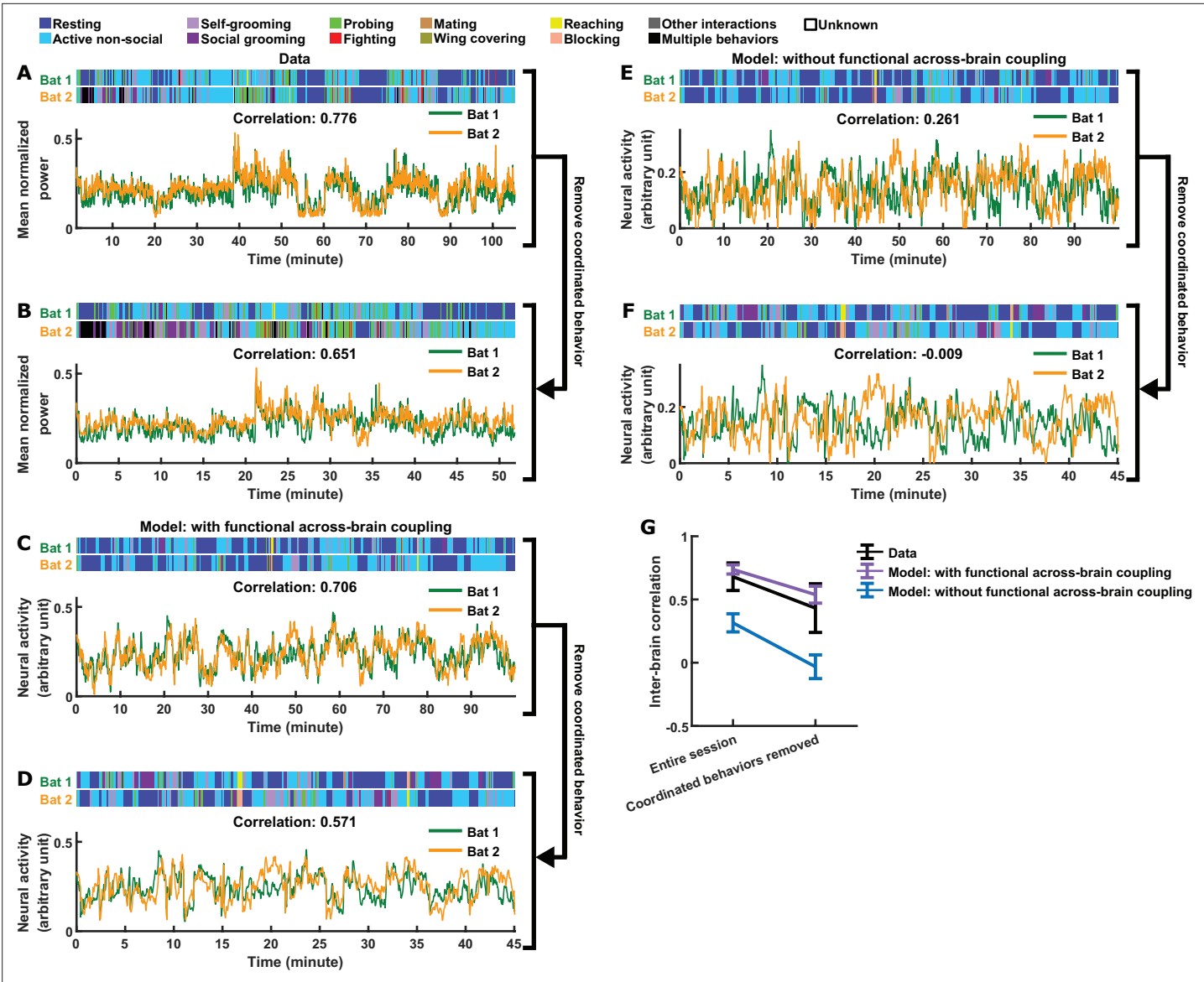

**Figure 4.** Comparing models with and without functional across-brain coupling. (**A**) Mean normalized LFP power in the 30–150 Hz band, averaged across all channels for each bat, on an example one-chamber session. Shown above are the annotated behaviors. Note that the activity of the two bats are highly correlated (correlation coefficient indicated above the activity traces). (**B**) The same data as in (**A**), after removing all time periods of coordinated behavior (i.e., time periods when both bats engaged in the same behavior). Inter-brain correlation was re-calculated and indicated above the activity traces. Note that the inter-brain correlation is lower than in (**A**), but still highly positive. (**C**)-(**D**) Same as (**A**)-(**B**), but for simulated neural activity and behaviors from an example one-chamber simulation of our main model. Note that inter-brain correlation remained after removing time periods of coordinated behavior, reproducing the experimental result. (**E**)-(**F**) Same as (**C**)-(**D**), but simulated without functional across-brain coupling. Specifically, the same simulated behavior and behavioral modulation $b(t)$ was used as in (**C**)-(**D**); the difference from (**C**)-(**D**) is that the strength of functional across-brain coupling $C_I$ was set to 0. Note that inter-brain correlation disappeared after removing time periods of coordinated behavior, unlike the experimental result. (**G**) Inter-brain correlation over entire sessions or after removing coordinated behavior, for data (black; mean normalized LFP power in the 30–150 Hz band), the main model (purple; with functional across-brain coupling), and the model without functional across-brain coupling (blue). The simulations of the two models were done in pairs: each simulation of the model without functional across-brain coupling used the same behavioral modulation (including behavioral modulation noise) as one simulation of the main model, as done in (**C**)-(**F**). Plotted are averages (± standard deviation) across experimental sessions or simulations. Note that functional across-brain coupling is required in the model to reproduce persisting inter-brain correlation after the removal of coordinated behaviors. See Materials and methods for details.

The online version of this article includes the following figure supplement(s) for figure 4:

**Figure supplement 1.** A Kuramoto model of the inter-brain relationship.

inter-brain correlation and the relative magnitudes of the mean and difference components from the data; importantly, however, it does not reproduce the relative timescales of the mean and difference components from the data (*Figure 4—figure supplement 1*). This demonstrates that not all forms of functional coupling are equally able of reproducing the inter-brain relationship.

In summary, our main model provides a simple explanation for a set of robust, yet puzzling relationships between the inter-brain mean and difference components. Namely, through opposite feedback to the mean and difference components, functional across-brain coupling simultaneously modulates both their magnitude and timescales in opposite directions—the amplification and slowing down of the mean and the suppression and speeding up of the difference are all manifestations of a single mechanism.

## Testing predictions emerging from the model

In addition to explaining experimental observations, the model also makes novel predictions about previously unexamined aspects of the inter-brain relationship. In this section, we describe three predictions about disparate aspects of the inter-brain relationship: (1) correlation between neural activity variables defined by rotations of the two-bat activity space axes; (2) relationship between the difference and mean components across sessions, rather than within sessions; (3) the encoding of the behaviors of one bat by the neural activity of the other bat. While the three predictions are seemingly unrelated to one another, they are in fact all motivated by the same model mechanism of functional across-brain coupling. After describing each prediction, we turn back to the data to test it. If the data can confirm the predictions, it would support the validity of the model.

In the model, the neural activity of each bat is a variable. These two activity variables can be thought of as evolving within a two-dimensional space, that is, a space whose axes are the neural activity of each of the two bats (*Figure 3G* left). The evolution of activity in this space can be equivalently described using alternative sets of activity variables: for example, by rotating the axes by 45°, the activity variables become the difference and mean components (*Figure 3G* right). Rotating the axes in this way changes not only the activity variables, but also the strength of functional coupling between the variables. In the one-chamber model, if we rotate the axes smoothly, the strength of functional coupling also changes smoothly (*Figure 5A*). In particular, the coupling strength is positive and at its maximum when the axes correspond to the activity of each bat, and it decreases to zero as the axes rotate by 45° (corresponding to the mean and difference components), and finally becoming negative as the axes rotate further (*Figure 5A*). After any amount of rotation, we can calculate the correlation between the activity variables. The model predicts that, as the axes rotate, the correlation between the neural activity variables would mirror the coupling strength (*Figure 5B*). This effect is not due to the changing behavioral modulation upon axes rotation, as can be seen after regressing out the influence of behavior from the activity (brown curve in *Figure 5B*; see Materials and methods for procedures used to regress out behavior). As we turn back to the data, we found that it clearly confirmed the model predictions: correlation as a function of rotation showed the same relationship as in the model (*Figure 5C*). Similarly, data from the two-chambers sessions (*Figure 5F*) also confirmed the predictions of the two-chambers model (*Figure 5D–E*): namely, near zero correlation across all axes rotation angles.

In the previous sections, we have examined in detail the relationships between the magnitudes and timescales of the inter-brain activity components in single sessions, but the model also makes predictions regarding their relationship across sessions. As described above and in Materials and methods, functional across-brain coupling acts as positive feedback for the mean component and negative feedback for the difference component, modulating both their magnitudes and timescales in opposite directions (*Figure 3H*). Thus, the model predicts that the relative magnitudes and relative timescales of the two components are tied together by this mechanism and do not vary independently. This can be seen by plotting the relative magnitudes against the relative timescales for different simulations, which shows linear relationships between them (*Figure 5G*; see Materials and methods for detailed analysis of these relationships). Turning to the data, we see similar linear relationships, again confirming the predictions of the model (*Figure 5H*). It is important to note that these are very specific predictions that were not at all obvious or expected from our previous knowledge of the inter-brain relationship. The fact that they were verified by the data provides strong support for the validity of the model.

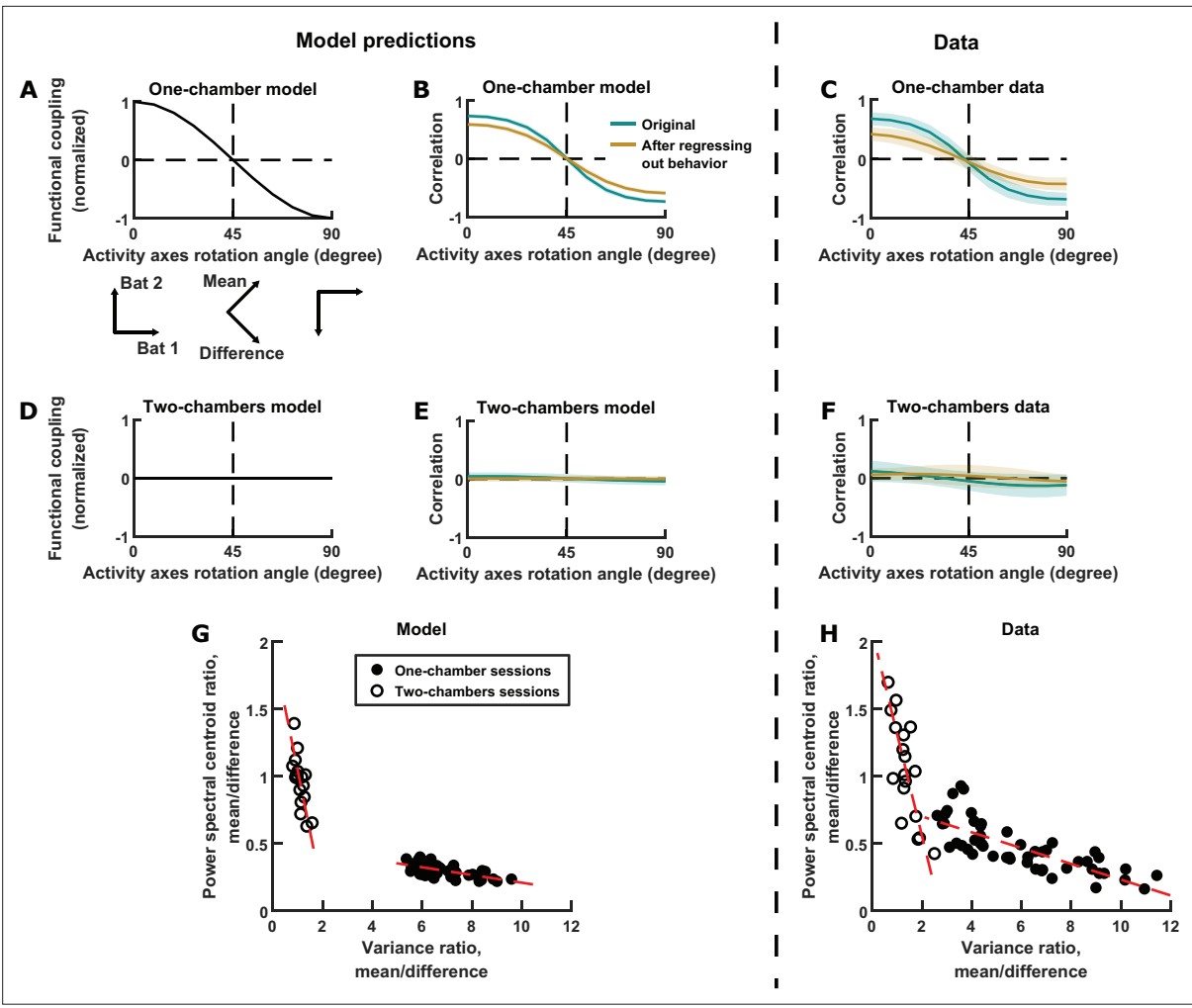

**Figure 5.** Testing model predictions. (**A**), (**B**), (**D**), (**E**), and (**G**) show model results and predictions, and (**C**), (**F**), and (**H**) show the corresponding data. (**A**) The neural activity of two bats can be represented in a 2D space, using any choice of axes orientation for the space. In our model, the axes orientation determines the strength of coupling between the activity variables represented by the axes. Here, functional coupling between the activity variables in the one-chamber model is plotted against the axes orientation defining the activity variables. When the activity variables correspond to the activity of each bat (0° rotation), functional coupling is at its positive maximum. Functional coupling decreases as the axes rotate, becoming zero when the activity variables correspond to the mean and difference components (45° rotation). Three orientations of the axes are illustrated below the plot. The plotted functional coupling was normalized by the functional coupling at 0° rotation. (**B**) Correlation between the activity variables in the one-chamber model, plotted against the orientation of the axes of the activity space that defines the activity variables. Correlations were shown both before (teal) and after (brown) regressing out the behaviors of both bats from each activity variable. Shading indicates standard deviation across simulations. Note that the shape of the correlation curves mirrors that of functional coupling shown in (**A**), and that regressing out behaviors changes the magnitude, but not the shape of the correlation curve. (**C**) Same as (**B**), but for data from one-chamber sessions. Shading indicates standard deviation across sessions. Note that the data confirms the model predictions from (**B**). (**D**)-(**F**) Same as (**A**)-(**C**), but for the two-chambers model and data. Note that the data again confirms model predictions. (**G**) Scatter plot of power spectral centroid ratio (mean component centroid divided by difference component centroid) vs. variance ratio (mean component variance divided by difference component variance). Each circle is a simulation (filled: one-chamber model; open: two-chambers model). Red dashed lines are total least squares regression lines. Note that the model predicts linear relationships between the variance ratio and centroid ratio. See *Figure 5—figure supplement 1* and Materials and methods for detailed analysis of these relationships. (**H**) Same as (**G**), but for the data. Note that the prediction from (**G**) is confirmed by the data.

The online version of this article includes the following figure supplement(s) for figure 5:

**Figure supplement 1.** Analysis of linear relationship between power spectral centroid ratio and variance ratio.

In the model, the neural activity of each bat is directly modulated by its own behavior; moreover, it is indirectly modulated by the behavior of the other bat, through functional across-brain coupling. This suggests that the neural activity of each bat should represent the behavior of the other bat independently from encoding its own behavior. To quantify this in the model, we first regressed out

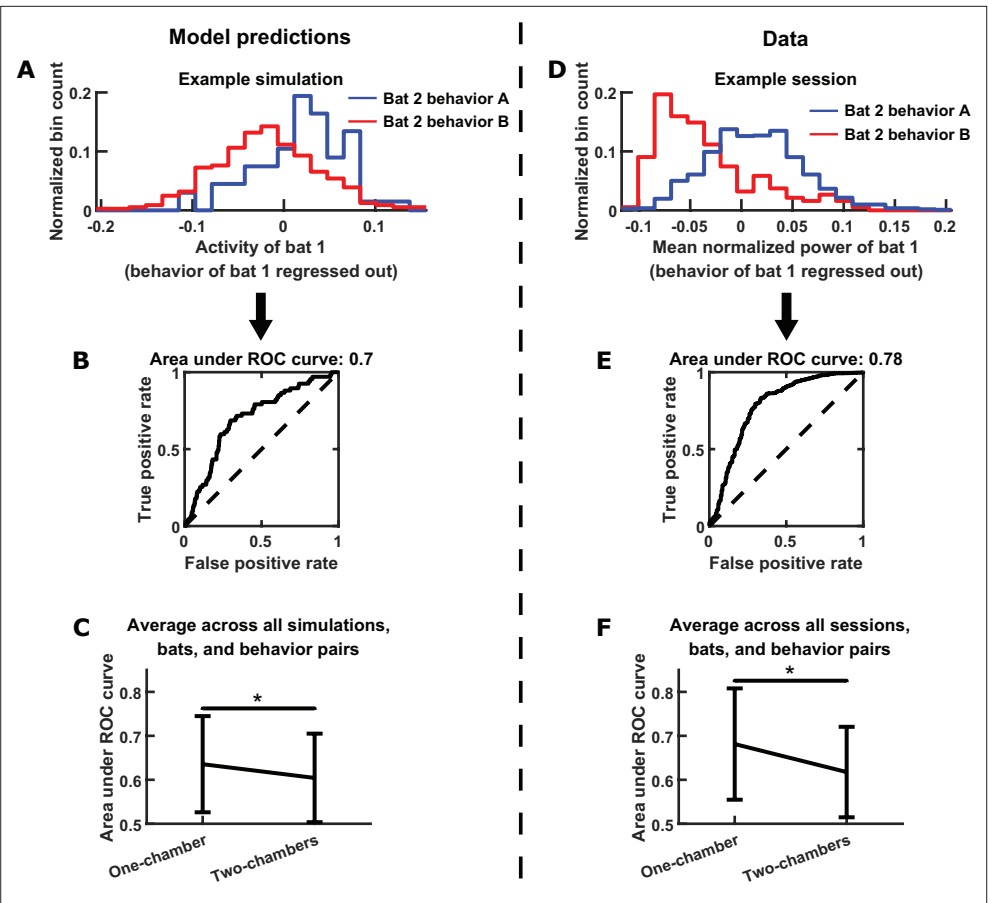

**Figure 6.** Testing further model predictions. (**A**)-(**C**) show model results and predictions, and (**D**)-(**F**) show the corresponding data. (**A**) Distributions of the neural activity of bat 1 conditioned on the behavior of bat 2. Bat 1's behavior has been regressed out of its neural activity. Two distributions are shown from an example simulation, for two example behaviors by bat 2 (behavior A: probing; behavior B: resting). Note that bat 1's activity encodes bat 2's behavior independently of its own behavior. (**B**) The extent to which bat 1's activity encodes the two behaviors of bat 2 in the example from (**A**) is quantified using an ROC curve, which illustrates the discriminability of the two distributions from (**A**). The area under the ROC curve is indicated above the plot (larger area indicates better discriminability). (**C**) Area under the ROC curve is averaged across all simulations, bats, and behavior pairs, separately for one-chamber simulations and two-chambers simulations. When using one bat's activity to discriminate the other bat's behavior, discriminability is significantly higher in one-chamber simulations. Error bars denote standard deviations. *, p < 0.05, Wilcoxon rank sum test. (**D**)-(**F**) Same as (**A**)-(**C**), but for mean normalized 30–150 Hz LFP power from the data. For the example in (**D**), behaviors A and B are active non-social and social grooming, respectively. Note that the data is consistent with model predictions.

the behavior of each bat from its own neural activity, then examined to what extent the residual neural activity encodes the behavior of the other bat, using the receiver operating characteristic (ROC) curve. The reason we first regress out each bat's behavior is the following. If two bats happened to engage in the same behavior at the same times, the activity of each bat would appear to encode the behavior of the other bat, when it was simply encoding its own behavior. By regressing out each bat's own behavior, this approach eliminates such potential spurious correlation between one bat's activity and another bat's behavior caused by coordinated behaviors between the bats. *Figure 6A* shows an example of one bat's activity encoding the other bat's behavior independently of its own behavior, and the strength of the encoding was quantified using the ROC curve in *Figure 6B*. Across simulations, we found that the encoding of the other bat's behavior was significantly stronger for the one-chamber model than the two-chambers model (*Figure 6C*), consistent with the presence of functional across-brain coupling in the one-chamber model. Applying the same approach to the data, we saw similar neural encoding of the other bat's behavior (example in *Figure 6D*, corresponding ROC curve

in *Figure 6E*). Across all data sessions, the encoding was stronger on one-chamber sessions than two-chambers sessions, confirming the model predictions (*Figure 6F*).

In summary, we tested and confirmed three disparate predictions of the model that all stem from the model mechanism of functional across-brain coupling. The fact that three seemingly unrelated experimental observations can all be explained by our model underscores its explanatory power.

## Modeling inter-brain neural relationship during group social interactions

So far, our analysis has been exclusively focused on social interactions between two individuals. However, social interactions also occur among larger groups in many species, including the group-living Egyptian fruit bats (*Herzig-Straschil and Robinson, 1978*; *Kwiecinski and Griffiths, 1999*). Although the inter-brain relationship between two interacting individuals have been studied extensively, the inter-brain relationship among larger social groups have received far less attention (but see *Dikker et al., 2017*; *Rose et al., 2021*). While there are many open questions regarding the group inter-brain relationship, in the context of the current study, we ask two questions that are natural extensions of our two-bat findings. First, can the opposite feedback mechanism be generalized to the larger group context? Second, what would such a mechanism predict about the group inter-brain relationship? To answer these questions, we extended our model to larger groups, which offered direct predictions regarding the group inter-brain relationship. We then tested these predictions using social group data in bats.

To extend the two-bat model to interactions among a group of $n$ bats, we used the same equation shown in *Figure 3A*, except that the activity $\boldsymbol{a}$ and behavioral modulation $\boldsymbol{b}$ are now $n$-dimensional rather than 2-dimensional, and the functional coupling matrix $\boldsymbol{C}$ is similarly $n \times n$. As in the two-bat model, the activity of each bat in the $n$-bat model is influenced by the same functional self-coupling $-\boldsymbol{C}_S$, and the activity of each pair of bats are functionally coupled with the same positive coupling $\boldsymbol{C}_I$.

The mechanism that governs the two-bat model naturally generalizes to the $n$-bat model (see Materials and methods for details). For the $n$-bat model, the mean component corresponds to the mean activity across all bats. On the other hand, a difference subspace takes the place of the difference component of the two-bat model. In $n$-bat activity space, the difference subspace is the $(n-1)$-dimensional subspace orthogonal to the direction of the mean component (*Figure 7B*). This subspace contains all inter-brain activity patterns that correspond to activity differences across any of the brains. Similar to the two-bat model, functional across-brain coupling acts as positive and negative feedback to the $n$-bat mean component and the difference subspace, respectively, amplifying and slowing down the $n$-bat mean component while suppressing and speeding up activity patterns in the difference subspace.

The opposite feedback mechanism in the $n$-bat model thus gives rise to predictions that can be tested experimentally. Specifically, one prediction is that activity patterns in the difference subspace would have smaller magnitudes than the $n$-bat mean component: as we show using simulations of a 4-bat model, the average variance of the difference subspace is smaller than the variance of the 4-bat mean component (*Figure 7C*). Another prediction is that activity patterns in the difference subspace would have faster timescales than the $n$-bat mean component: 4-bat simulations show that the difference subspace has a higher average power spectral centroid than the 4-bat mean component (*Figure 7D*). Finally, similar to the two-bat model (see *Figure 5A–B*), correlation between activity variables depends on their functional coupling. In particular, the positive functional across-brain coupling between the activity of different bats would give rise to positive inter-brain correlations for each pair of bats within the group; on the other hand, the mean component is not functionally coupled with activity patterns in the difference subspace, so they would therefore be uncorrelated. This prediction is illustrated in *Figure 7E* using simulations of a 4-bat model.

Next, we tested these predictions using experimental data collected from a group of socially interacting bats. In this data set, neural activity was simultaneously and wirelessly recorded from the frontal cortices of four bats while they interacted in a chamber, under the same condition as the one-chamber sessions described above (*Rose et al., 2021*). As with the two-bat experiment, we focused on LFP power in the 30–150 Hz band. Applying the same analyses used for the 4-bat model simulations in *Figure 7C–E*, we found that the data qualitatively confirmed all model predictions: activity patterns in the difference subspace had smaller magnitudes (*Figure 7F*) and faster timescales (*Figure 7G*)

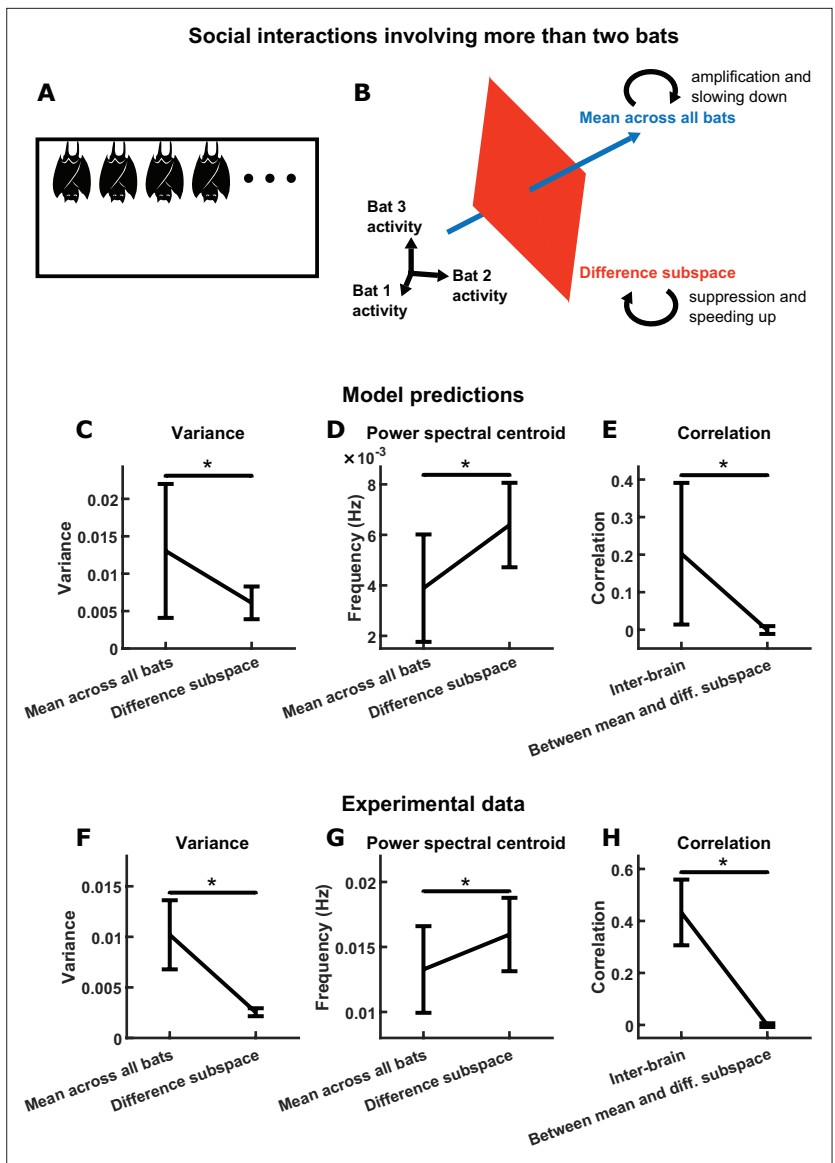

**Figure 7.** Group inter-brain relationship: model predictions and experimental tests. (**A**) The two-bat model can be generalized to model the neural activity of a group of more than two socially interacting bats (the $n$-bat model). (**B**) The mechanism underlying the $n$-bat model, illustrated for the case of $n = 3$ bats (three was chosen here only because higher-dimensional spaces cannot be illustrated). Shown are the mean component and the difference subspace in the activity space of $n$ bats. The mean component, corresponding to the mean activity across all bats, is a vector pointing towards the first octant. The difference subspace is the $(n - 1)$-dimensional subspace orthogonal to the direction of the mean component, and it contains all inter-brain activity patterns that correspond to activity differences across brains. Similar to the two-bat model, functional across-brain coupling acts as positive feedback to the $n$-bat mean component, amplifying it and slowing it down; and as negative feedback to activity patterns in the difference subspace, suppressing them and speeding them up. (**C**)-(**E**) Model predictions from simulations of an $n$-bat model where $n = 4$. Plotted are averages (± standard deviation) across simulations. (**C**) Model predictions on the variance of the mean component and the average variance of the difference subspace (total variance in the difference subspace divided by $n - 1$). (**D**) Model predictions on the power spectral centroid of the mean component and the average power spectral centroid of the difference subspace (averaged across the power spectral centroids of activity in 1000 random directions in the difference subspace for a given simulation). (**E**) The model predicts positive inter-brain correlation and zero average correlation between the mean component and activity patterns in the difference subspace. Here, for each simulation, inter-brain correlation was averaged across all pairs of bats, and average correlation was calculated between the mean component and activity in 1000 random directions in the difference subspace. (**F**)-(**H**) Experimental tests. Neural activity was

*Figure 7 continued on next page*

*Figure 7 continued*

simultaneously recorded from the frontal cortices of four bats while they interacted in a chamber, under the same condition as the one-chamber sessions from *Figure 1A*. Mean normalized 30–150 Hz LFP power from the four bats were analyzed using the same methods as in (**C**)-(**E**). Note that the experimental results confirm the qualitative trends from the model predictions (although the data differ quantitatively from the predictions; see Materials and methods). *, p < 0.05, Wilcoxon signed rank test.

than the 4-bat mean component, there was positive inter-brain correlation between bats in the group (*Figure 7H*), and there was no correlation between the 4-bat mean component and activity patterns in the difference subspace (*Figure 7H*). Note that the 4-bat experimental data and the model predictions differ quantitatively, because the 4-bat model did not include realistic behavioral modeling as did the two-bat model (Materials and methods).

In conclusion, we modeled the group inter-brain relationship and experimentally confirmed the model predictions. Through this, we found that the opposite feedback mechanism is not restricted to pair social interactions, but naturally generalizes to larger social groups.

## Discussion

Here we studied the inter-brain relationship between socially interacting bats, by analyzing the mean and difference between the bats' neural activity. We found that the difference in activity has smaller magnitudes and faster timescales than the mean, which is a robust neural signature of social interactions. We reproduced this finding using a simple model, which suggests a specific computational mechanism shaping the inter-brain neural relationship: functional coupling across brains acting as opposite feedback to the difference and mean activity components. This mechanism gave rise to a number of predictions, which we then confirmed using experimental data from bats engaging in paired and group social interactions. Furthermore, this mechanism has broad explanatory power, and can account for a range of previously observed features of the inter-brain neural relationship, as we discuss below.

First, it is well known that neural activity is correlated between the brains of socially interacting individuals, in mice, bats, and humans (e.g. *Dikker et al., 2014*; *Kingsbury et al., 2019*; *Kinreich et al., 2017*; *Levy et al., 2017*; *Montague et al., 2002*; *Piazza et al., 2020*; *Spiegelhalder et al., 2014*; *Stolk et al., 2014*; *Zadbood et al., 2017*; *Zhang and Yartsev, 2019*). Through negative feedback to the difference component, functional across-brain coupling suppresses the inter-brain difference and keeps it small. At the same time, positive feedback to the mean component amplifies the common activity covariation between brains, so that the neural activity in each is dominated by their common activity pattern. Together, these would result in the inter-brain correlation observed during social interactions in diverse species.

Second, in a shared social environment, neural correlation across brains is above and beyond what would be expected from behavioral coordination between individuals (*Kingsbury et al., 2019*; *Piazza et al., 2020*; *Zhang and Yartsev, 2019*). This can be understood because, through opposite feedback to the difference and mean components, functional across-brain coupling contributes to correlation independently from behavioral coordination. As a consequence of this, even in the absence of behavioral coordination, the opposite feedback results in correlation across brains in the model (*Figure 4*), consistent with previous experimental observations (*Kingsbury et al., 2019*; *Piazza et al., 2020*; *Zhang and Yartsev, 2019*).

Third, the neural correlation between socially interacting individuals varies as a function of timescale: correlation across brains is higher for activity at slower timescales, and lower for activity at faster timescales (*Kingsbury et al., 2019*; *Zhang and Yartsev, 2019*). This would be a natural consequence of the opposite feedback mechanism. Positive feedback to the mean amplifies it and slows it down, while negative feedback to the difference suppresses it and speeds it up. Thus, at slower timescales, the activity of the two brains is dominated by the mean component, that is, the activity of the two brains are very similar at slower timescales and thus have higher correlations. On the other hand, at faster timescales, the increasing presence of the difference component lowers the correlation across brains.

In summary, our model offers a particularly simple explanation for diverse aspects of the inter-brain neural relationship during both pair and group social interactions, including: (1) neural correlation across brains and the relative magnitudes of inter-brain mean and difference components (*Figure 3E*; *Figure 7F and H*); (2) neural correlation beyond behavioral coordination; (3) neural correlation as a function of timescale; (4) the relative timescales of inter-brain mean and difference components (*Figure 3F*; *Figure 7G*); (5) correlation between activity variables under rotated activity bases (*Figure 5B–C*); (6) the relationship between the magnitudes and timescales of activity components across sessions (*Figure 5G–H*); and (7) modulation of one bat's neural activity by the behavior of another bat (*Figure 6*). Importantly, there is no reason a priori to suspect that these disparate observations are all related; only in light of the model is it apparent that they are all manifestations of a single underlying computational mechanism. Collectively, these results thus suggest opposite feedback to the difference and mean as a unifying computational mechanism governing the inter-brain relationship during social interactions. This points to a major avenue of future research: elucidating the biological implementation of this computational mechanism. Such research will likely involve studying the distributed neural circuits mediating sensorimotor interactions and attentional processes during social interactions, to understand how they might functionally act as opposite feedback to the inter-brain difference and mean components. To do so, a relevant experiment is to record from multiple brain regions and examine how signals related to social interactions are transformed across regions. This will also reveal the extent to which inter-brain correlation varies across regions, allowing comparison to the human literature, where variations across brain structures have long been studied using EEG and fMRI.

Another important future direction is comparative studies of the inter-brain relationship to elucidate similarity and differences across species. So far, this has been studied in three non-human species: bats, mice (*Kingsbury et al., 2019*), and birds (*Hoffmann et al., 2019*). While bats and mice showed qualitatively similar inter-brain relationships, birds were different, showing anti-correlated neural activity across brains during a duetting behavior (*Hoffmann et al., 2019*). It would be interesting to study the inter-brain relationship in species with different levels of sociality and different types of social behaviors, for exmple, solitary animals that only socially interact for mating, or animals that live in large groups. Such comparative studies, with careful behavioral quantification, would identify neural mechanisms that underlie social interactions in general versus species-specific social behaviors. Moreover, the analytic and modeling methods used here can be easily applied to other species, enabling comparison across species under the same framework. In particular, studying the timescales of inter-brain activity patterns during different social behaviors in different species could lead to a more general understanding of functional across-brain coupling.

Further future directions are motivated by the limitations of our model. For simplicity, our model took functional across-brain coupling to be static, that is, taking the same functional form and having the same strength independent of the bats' behaviors. In reality, the coupling is mediated through behavioral interaction between the bats. Thus, it is likely dynamic, taking different forms and varying in strength depending on the bats' dynamic behaviors. For example, when one bat is actively grooming another passively resting bat, functional coupling might be asymmetric between them, stronger in one direction than the other. Conversely, when two bats are fighting with each other, their functional coupling might be more symmetric. Furthermore, the timescales of functional coupling could also vary: different behaviors might couple neural activity with different time courses and delays. Future study focusing on specific behaviors (e.g. grooming, fighting, etc.), with fine-timescale behavioral tracking (*Mathis et al., 2018*; *Pereira et al., 2019*), will be needed to identify the detailed behavior-specific dynamics of functional across-brain coupling.

Another limitation of our model is the 'open-loop' nature of the relationship between behavior and neural activity. Specifically, we modeled neural activity as being modulated by behavior, but behavior was modeled using a Markov chain that is independent from the neural activity. In reality, neural activity and behavior form a closed-loop, with different social behaviors being controlled by the neural activity of specific neural populations in specific brain regions. Thus, an important future direction is to close the loop by incorporating neural control of social behaviors into models of the inter-brain relationship in bats. This will require future experimental studies to identify which frontal cortical regions and populations in bats are necessary or sufficient to control social behaviors, as well as the detailed causal relationship from neural activity to social behavior. Furthermore, as social interactions can occur

at multiple timescales, it will be interesting to investigate how these are controlled by neural activity at different timescales, and how those timescales are shaped by functional across-brain coupling. In summary, such a closed-loop model will shed light on how inter-brain activity patterns and dynamic social interactions co-evolve and feedback onto each other.

# Materials and methods

## Request for resources
Further information and requests for resources should be directed to Michael M. Yartsev (myartsev@berkeley.edu).

## Experimental and data processing methods
The data analyzed in this study have been previously published in *Zhang and Yartsev, 2019* and *Rose et al., 2021*. The details of experimental methods and data processing methods for the two-bat social interaction experiments were published in *Zhang and Yartsev, 2019*; the details for the group social interaction experiments were published in *Rose et al., 2021*; therefore, here we present a brief summary of the published methods. All data processing was performed using MATLAB (MathWorks).

### Experimental subjects
Data was collected from eight adult male Egyptian fruit bats (*Rousettus aegyptiacus*). All procedures were approved by the Animal Care and Use Committee of UC Berkeley.

### Experimental setup
Experiments were conducted inside 40.6 × 33.7 × 52.1 cm (length × width × height) cages, placed in dark chambers. High-speed infrared video cameras (Flea3 or Chameleon3, FLIR) were used to record videos of the bats, and ultrasonic microphones (USG Electret Ultrasound Microphone, Avisoft Bioacoustics) were used to record audio.

The experiments included one-chamber sessions and two-chambers sessions. In the one-chamber sessions, bats behaved freely and interacted with each other inside a chamber. In the two-chambers sessions, bats behaved freely in separate, identical chambers. There were three types of two-chambers sessions: (1) two bats each freely behaving in isolation; (2) two bats each freely behaving in the presence of identical auditory stimuli (playback of bat calls); (3) two bats each freely behaving and interacting with a different partner in separate chambers.

### Behavior annotation
For the two-bat experiments, the behaviors of the bats were manually annotated by experienced observers. The annotated behaviors and their definitions are as follows.

- *Resting*. A bat hanging by its feet, with its head and body still.
- *Active non-social*. A bat engaging in any kind of active behavior that does not involve social interaction, including: the bat hanging by its feet or feet and thumbs, and moving its head or body; the bat climbing or crawling around; the bat shaking its body; the bat jumping or flying off from the roof of the cage.
- *Self-grooming*. A bat either licking or scratching itself.
- *Social grooming*. A bat either licking or scratching another bat.
- *Probing*. A bat poking its snout at the head or body of another bat.
- *Fighting*. A bat moving its wings or thumbs to quickly hit another bat, or biting another bat.
- *Mating*. A male bat inserting or attempting to insert its penis into a female bat's vagina.
- *Wing covering*. A bat struggling with another bat in order to cover the other bat's body with its opened wings.
- *Reaching*. A bat attempting to reach over the body or wings of another bat with its head or thumbs.
- *Blocking*. A bat using its wings to actively block another bat from accessing a location.
- *Other interactions*. Any social interaction other than the ones already defined.

## Surgery

Anesthesia and surgery were performed as described previously in *Yartsev and Ulanovsky, 2013* to implant the bats with four-tetrode microdrives (Harlan 4 Drive, Neuralynx). To target the frontal cortex, the center of craniotomy was positioned at 1.7 mm lateral to the midline and 12.19 mm anterior to the transverse sinus located between cortex and cerebellum.

## Electrophysiological recording

Electrophysiology was performed using a wireless neural data logging system (Neurolog-16 and MouseLog-16, Deuteron Technologies). Tetrodes were advanced ventrally every one to two days (mostly by 20–160 μm), to record activity at different sites.

## Histology

To determine the neural recording sites, we performed histology as previously described (*Yartsev and Ulanovsky, 2013*), using Nissl staining.

## Preprocessing of electrophysiological data

To obtain LFP, we low-pass filtered the raw voltage traces (cut-off frequency: 200 Hz), and then downs-ampled it to 496.6 Hz (the two-bat experiments) or 520.8 Hz (the four-bat experiments).

We detected spikes from band-pass filtered (600–6000 Hz) voltage traces using threshold crossing. Spike sorting was done automatically using SNAP Sorter (Neuralynx), then manually checked using SpikeSort3D (Neuralynx). For each tetrode on each session, all spikes not assigned to single units were grouped into a multiunit. All units with firing rate below 2 Hz were excluded from further analysis.

## Calculation of LFP spectrograms

To calculate LFP spectrograms, power spectra were calculated using the multitaper method for 5 s sliding windows of the LFP trace, with 2.5 s overlap between consecutive windows. To analyze different frequencies of the LFP on equal footing, we separately peak-normalized the power at each frequency for each spectrogram.

## Calculation of firing rates

Firing rates for single units and multiunits were computed in 5 s bins with 2.5 s overlap between consecutive bins.

## Dimensionality reduction of LFP

Previously, we found that the spectrograms of bat frontal cortical LFP can be reduced to two signals, power in the 1–29 Hz band and the 30–150 Hz band (*Zhang and Yartsev, 2019*). To analyze these signals, for a given normalized LFP spectrogram (with power peak-normalized for each frequency), we averaged the normalized power values across 1–29 Hz and across 30–150 Hz at each time bin. This is the 'mean normalized LFP power' used for all analyses in this paper involving LFP power (e.g. *Figure 1C*).

## Analysis and modeling

All analysis and modeling were performed using MATLAB (MathWorks).

## Statistical tests

The statistical tests used are stated in the figure legends. A significance level of 0.05 was used for all tests. Tests were two-tailed unless otherwise indicated.

## Inter-brain difference and mean components: definition and quantification

The neural activity of two bats can be represented as a two-dimensional vector $\begin{pmatrix} a_1(t) \\ a_2(t) \end{pmatrix}$, where $a_1(t)$ and $a_2(t)$ are the neural activity (normalized LFP power, multiunit activity, or single unit activity) of bat 1 and bat 2 at time $t$, respectively. Through a change of basis, the same activity can be

represented under another orthogonal basis as the mean and difference between the two brains:

$\begin{pmatrix} a_M(t) \\ a_D(t) \end{pmatrix} = \frac{1}{2} \begin{pmatrix} a_1(t) + a_2(t) \\ a_1(t) - a_2(t) \end{pmatrix}$, where $a_M(t)$ is the mean component of the activity, and $a_D(t)$ is

the difference component. Here, the difference component is defined as $\frac{1}{2}\left[a_1(t) - a_2(t)\right]$ rather than

$a_1(t) - a_2(t)$ so as to have the same scale as the mean component. $a_M(t)$ represents the common activity pattern shared between the brains, and $a_D(t)$ represents the activity difference between the brains.

To quantify the magnitude of an activity component on a given session, we computed the variance of its time series over the session. To quantify the timescales of an activity component on a given session, we computed its power spectral centroid as follows. Given the time series of an activity component over the session, we subtracted from it its average over time, multiplied it with a Hamming window, and then computed the periodogram estimate of its power spectrum. The power spectral centroid is then computed as a weighted average of frequency, with the power at each frequency as its weight. A higher power spectral centroid means that the activity has more power at higher frequencies, so that the time series varied at faster timescales. Note that the power spectral centroid was always calculated from time series of mean normalized LFP power or firing rate, and not from time series of LFP itself.

## Surrogate data and the relationship between inter-brain correlation and mean and difference components

In our data from one-chamber sessions, the mean component had large magnitudes and slow timescales, and the difference component had small magnitudes and fast timescales. The relative magnitudes of the two components are expected, given that activity from the two bats showed similar patterns over time (*Figure 1C*), with high positive inter-brain correlations (*Zhang and Yartsev, 2019*); as explained below, a positive correlation mathematically implies larger variance for the mean component compared to the difference component. What about their timescales? Are the observed relative timescales of the two components necessary mathematical consequences of high inter-brain correlations, large mean components, and small difference components?

To answer this, we examine the relationship between inter-brain correlation and the mean and difference components. Let's use $A_1$, a column vector, to denote the activity of bat 1 during a session, and similarly, $A_2$ for the activity of bat 2. $A_1$ and $A_2$ are $N$-dimensional vectors, where $N$ is the number of time points in the session. The activity of the mean component is $A_M = (A_1 + A_2)/2$, and the activity of the difference component is $A_D = (A_1 - A_2)/2$. We use $\hat{A}_1$ to denote the mean-subtracted version of $A_1$ (here 'mean-subtracted' refers to subtracting the average across the elements of a vector, that is, across time, not to be confused with the average across bats, as in the 'mean component'), and similarly, $\hat{A}_2$, $\hat{A}_M$, and $\hat{A}_D$ for mean-subtracted versions of the respective vectors. Then, the Pearson correlation coefficient between the activity of two brains is:

$$
\begin{aligned}
correlation &= \frac{\hat{A}_1^T}{\sqrt{\hat{A}_1^T\hat{A}_1}} \frac{\hat{A}_2}{\sqrt{\hat{A}_2^T\hat{A}_2}} \\
&= \frac{(\hat{A}_M + \hat{A}_D)^T}{\sqrt{(\hat{A}_M + \hat{A}_D)^T(\hat{A}_M + \hat{A}_D)}} \frac{(\hat{A}_M - \hat{A}_D)}{\sqrt{(\hat{A}_M - \hat{A}_D)^T(\hat{A}_M - \hat{A}_D)}} \\
&= \frac{\hat{A}_M^T\hat{A}_M - \hat{A}_D^T\hat{A}_D}{\sqrt{(\hat{A}_M^T\hat{A}_M + \hat{A}_D^T\hat{A}_D)^2 - 4(\hat{A}_M^T\hat{A}_D)^2}}
\end{aligned}
\tag{1}
$$

Thus, the correlation depends on the quantities $\hat{A}_M^T\hat{A}_M$, $\hat{A}_D^T\hat{A}_D$, and $\left(\hat{A}_M^T\hat{A}_D\right)^2$. Note that $\hat{A}_M^T\hat{A}_M$ is $N$ times the variance of the mean component, and $\hat{A}_D^T\hat{A}_D$ is $N$ times the variance of the difference component. A positive correlation requires the numerator in *equation (1)* to be positive, in other words, the mean component having larger variance than the difference component, as stated above.

*Equation (1)* also shows that having a given combination of correlation, mean component variance, and difference component variance does not place constraints on the timescales of the mean and difference components. Specifically, it does not constrain the difference component to be faster than the mean. To explicitly demonstrate this, we generated surrogate data with identical inter-brain

correlation, mean component variance, and difference component variance as actual data, but having a slower difference component than the mean component (**Figure 2**).

The surrogate data in **Figure 2A–C** were generated from the actual 30–150 Hz LFP power data of the example session shown in **Figure 1C**, by keeping the mean component of the actual data, and replacing the difference component with a surrogate, using the following procedure. Below we denote the actual data by $A_D$, $\hat{A}_D$, etc. as above, and denote their surrogate counterparts by $S_D$, $\hat{S}_D$, etc. We generated a random $N \times 1$ activity vector by picking each element independently from the uniform distribution between 0 and 1, smoothed it with a 1000-second moving average filter, and subtracted from it the average across its elements. Let's call the resulting vector $\hat{R}$. Due to the smoothing, $\hat{R}$ varied on slow timescales. We then found the component of $\hat{R}$ that is orthogonal to $\hat{A}_M : \hat{R}_O = \hat{R} - [(\hat{R}^T\hat{A}_M)/(\hat{A}_M^T\hat{A}_M)]\hat{A}_M$. Next, we constructed a vector $\hat{S}'_D$ as a linear combination of $\hat{A}_M$ and $\hat{R}_O : \hat{S}'_D = \cot[\arccos(\hat{A}_M^T\hat{A}_D/\sqrt{\hat{A}_M^T\hat{A}_M\hat{A}_D^T\hat{A}_D})]\hat{A}_M/\sqrt{\hat{A}_M^T\hat{A}_M} + \hat{R}_O/\sqrt{\hat{R}_O^T\hat{R}_O}$. The surrogate for $\hat{A}_D$ is the vector $\hat{S}_D$, obtained by scaling $\hat{S}'_D$ to have the same vector norm as $\hat{A}_D : \hat{S}_D = (\sqrt{\hat{A}_D^T\hat{A}_D}/\sqrt{\hat{S}'^T_D\hat{S}'_D})\hat{S}'_D$. The surrogate difference component is then $S_D = \hat{S}_D + \bar{A}_D$, where every element of the vector $\bar{A}_D$ is equal to the average across the elements of $A_D$. The surrogate activity of bat 1 and bat 2 are, respectively, $S_1 = A_M + S_D$ and $S_2 = A_M - S_D$. This procedure ensures that $\hat{S}_D^T\hat{S}_D = \hat{A}_D^T\hat{A}_D$ and $\left(\hat{A}_M^T\hat{S}_D\right)^2 = \left(\hat{A}_M^T\hat{A}_D\right)^2$, so that the surrogate data had identical inter-brain correlation, mean component variance, and difference component variance as the actual data from which it was generated. Note that here we chose to leave the mean component $A_M$ from the actual data unchanged in generating the surrogate data, so that the surrogate $S_1$ and $S_2$ show qualitatively similar behavioral modulation over time as the actual data $A_1$ and $A_2$, but it is also possible to replace both $A_M$ and $A_D$ with surrogate counterparts.

For **Figure 2D–E**, we repeated the above procedure for each of the one-chamber sessions from the actual data set, using the actual 30–150 Hz LFP power data.

## Modeling: procedure

Our goal of modeling was to reproduce the observed relationship between difference and mean components using a model that is simple and parsimonious, in order to unambiguously identify the underlying computational mechanisms. We modeled the time evolution of the neural activity of two bats using the following differential equation:

$$\tau \frac{da(t)}{dt} = Ca(t) + b(t) \tag{2}$$

Here, $\tau$ is a time constant; $t$ is time; $a(t) = \begin{pmatrix} a_1(t) \\ a_2(t) \end{pmatrix}$ is the activity of bat 1 and bat 2 at time $t$; $C = \begin{pmatrix} -C_S & C_I \\ C_I & -C_S \end{pmatrix}$ is the functional coupling matrix, where $C_S$ is the strength of functional self-coupling and $C_I$ is the strength of functional across-brain coupling; $b(t) = \begin{pmatrix} b_1(t) \\ b_2(t) \end{pmatrix}$ is the strengths of behavioral modulation, where $b_1(t)$ is the modulation of the neural activity of bat 1 by the behavior of bat 1 at time $t$, and similarly for $b_2(t)$.

The functional across-brain coupling term, $C_I$, represents the indirect influence (as opposed to direct influence from an actual neural connection) one bat's neural activity has on the other bat's neural activity. For one-chamber sessions, $C_I > 0$, which models positive functional coupling when the two bats share a social environment (**Hasson et al., 2012**). For example, when bat 1's neural activity (say, 30–150 Hz LFP power) increases due to its active movements (**Gervasoni et al., 2004**; **McGinley et al., 2015**), the movements create sensory inputs to bat 2, which can increase bat 2's neural activity to the extent that bat 2 is paying attention (**Chun et al., 2011**; **Driver, 2001**; **Fritz et al., 2007**; **Reynolds and Chelazzi, 2004**). For two-chambers sessions, $C_I = 0$ since the two bats are separated. To ensure stability (so that neural activity do not go to infinity), $-C_S$ must be negative and must have a larger absolute value than $C_I$; thus, $0 \le C_I < C_S$.

To generate $b(t)$ for a simulation, we first simulated the bats' behaviors using a Markov chain. Each state of the Markov chain corresponds to the behaviors of the two bats at a given time: for example, bat 1 resting and bat 2 self-grooming would be one state. For the Markov chain for one-chamber

sessions, the transition probability matrix, initial distribution, and state space were determined as follows. The transition probability from one state to another is taken to be the empirical frequency of that transition during all one-chamber sessions. To calculate the empirical frequency, the behavior of each bat was sampled every 2.5 s, at the same time points as the neural activity (i.e. the center time point of each window used to calculate LFP power). A transition was counted for each consecutive pair of time points (in the rare instances where a bat engaged in multiple behaviors at the same time point, one transition was counted for each of the simultaneous behaviors). The initial distribution was taken to be the distribution of the behavioral state at the first time point of each one-chamber session. For these calculations, the two bats were assumed to be symmetrical, in the following sense. Using 'AB' to denote the state of 'bat 1 engaging in behavior A, bat 2 engaging in behavior B', we consider the transitions 'AB→CD' and 'BA→DC' to have the same transition probability. When calculating the empirical transition frequencies, the count for 'AB→CD' and the count for 'BA→DC' were each taken to be the sum of the actual counts for 'AB→CD' and 'BA→DC'. This applied also to states where both bats were engaging in the same behavior: for exmple, the count for the transition 'AA→AA' was taken to be double the actual count. This symmetry assumption was made for simplicity, and to allow behavioral data from different pairs of bats to be pooled together. Once the empirical transition frequencies were calculated, if there were less than 100 transitions from a given state, then that state was excluded from the state space. The same procedures were used to determine the transition probability matrix, initial distribution, and state space for the two-chambers Markov chain. The transition probability matrices for one-chamber and two-chambers sessions are shown in *Figure 3—figure supplement 2C-D*.

For each simulated session, we simulated the bats' behaviors for 100 minutes using the Markov chain (with 2.5 s between steps, the behavioral sampling period used when calculating the empirical transition frequencies). The first state of each simulated session was drawn randomly from the appropriate initial distribution. From the simulated behaviors, $\boldsymbol{b}(t)$ was determined at the discrete time points of the Markov chain steps (at integer multiples of 2.5 s). At the time point of a given Markov chain step, say $t_1$, $\boldsymbol{b}(t_1)$ was the sum of the noiseless deterministic behavioral modulation $\boldsymbol{b}_d(t_1)$ and the behavioral modulation noise $\boldsymbol{b}_n(t_1)$. $\boldsymbol{b}_d(t_1)$ depended on the behaviors of the two bats at time $t_1$. For example, if bat 1 was resting and bat 2 was engaging in self-grooming at time $t_1$, then $\boldsymbol{b}_d(t_1) = \begin{pmatrix} b_{resting} \\ b_{self-grooming} \end{pmatrix} + b_{constant}$, where $b_{resting}$ and $b_{self-grooming}$ are parameters specifying the level of behavioral modulation associated with the two behaviors, and $b_{constant}$ is a constant offset that differs between the one-chamber and two-chambers models, reflecting the effects of the general level of arousal that differs between the two conditions. See section 'Modeling: parameters' below for the parameter values for all the behaviors and $b_{constant}$. For the behavioral modulation noise, the two elements (for the two bats) of $\boldsymbol{b}_n(t_1)$ was each drawn independently from a Gaussian distribution with zero mean and standard deviation $\sigma_n$. After determining $\boldsymbol{b}(t) = \boldsymbol{b}_d(t) + \boldsymbol{b}_n(t)$ at the discrete time points of the Markov chain steps, $\boldsymbol{b}(t)$ at time points in-between were linearly interpolated.

Having simulated the behaviors and generated $\boldsymbol{b}(t)$ for a 100 minute session, we set the initial condition $\boldsymbol{a}(0)$ to be the fixed point under the initial noiseless behavioral modulation $\boldsymbol{b}_d(0)$: $\boldsymbol{a}(0) = -\boldsymbol{C}^{-1}\boldsymbol{b}_d(0)$. Then, *equation (2)* was numerically integrated (*ode45* function in MATLAB) to simulate neural activity $\boldsymbol{a}(t)$. The only differences between simulations of one-chamber and two-chambers sessions were the value of $\boldsymbol{C}_I$ and the behavior transition probability matrix. For analyses and figures involving $\boldsymbol{a}(t)$, $\boldsymbol{b}(t)$, $\boldsymbol{b}_d(t)$, and $\boldsymbol{b}_n(t)$ (*Figures 3–7* and their figure supplements), we used their values taken at discrete time points corresponding to the Markov chain steps (i.e. same sampling period as the data).

## Modeling: comparing behavior models

As described in the previous section, we modeled the behavior of the bats using a Markov chain. To justify this choice, we tested the Markov assumption by comparing three behavior models, including the Markov chain model we used. The first model is the independent model, where the behavioral state at a given time point is independent from the state at other time points. The second model is the first-order dependency model, where the behavioral state at a given time point depends on the state at the previous time point only; this was implemented as a part of our main model in the previous section, and it corresponds to the Markov assumption. The third model is the second-order

dependency model, where the behavioral state at a given time point depends on the states at the two previous time points. Models with longer time-dependencies ($\geq 3$) were not tested because the number of parameters grows exponentially with model order and cannot be fitted with our dataset.

The three models were each fitted separately for the one-chamber and two-chambers sessions. Each type of sessions was split into a training set and a test set of sessions (80% of the sessions for training and 20% for test). The models were fit by setting the probabilities or conditional probabilities of behaviors to the empirically observed frequencies (Laplace smoothing was used to avoid assigning zero probability to unobserved events). Then, the log-likelihood of the test set under each model was calculated. This procedure was repeated for 100 random splits of the data into training and test sets, and the log-likelihoods are shown in *Figure 3—figure supplement 2A, B*.

As shown in *Figure 3—figure supplement 2A, B*, the first-order model had the highest likelihood on average. This does not necessarily prove that bat behavior obeys the Markov assumption (e.g. given more data, it is possible that better second-order and higher order models could be fit). But it does mean that, given the amount of data we have, the best behavior model that we can fit is the first-order Markov chain, supporting its use in our main model described in the previous section.

## Modeling: comparing actual and simulated neural activity as a function of time

To compare neural activity as a function of time between the data and the model, we simulated our model using experimentally observed behaviors. In other words, rather than generating $b(t)$ from behaviors simulated using a Markov chain, we generated $b(t)$ based on the actual behaviors observed on a given experimental session. Neural activity was then simulated and compared to the actual neural activity on that experimental session. To quantify how well the model reproduces neural activity over time, we calculated the correlation between simulated and actual neural activity for each session and each bat: the average correlation over all sessions and bats is 0.72 (standard deviation 0.10). An example session is shown in *Figure 3—figure supplement 1*.

## Modeling: magnitudes and timescales of mean and difference components

How is the model able to reproduce the set of experimental observations relating the magnitudes and timescales of the mean and difference components (*Figure 3E–F*)? To understand the mechanisms in the model, we examine *equation (2)*. In general, in equations of this form, the activity variables are coupled: for example, in the one-chamber model, the activity of bat 1 $a_1(t)$ influences the activity of bat 2 $a_2(t)$, which in turn feeds back on $a_1(t)$. It is easier to understand how the activity evolves if we express the equation in a basis in which the activity variables uncouple. This basis consists of the eigenvectors of the functional coupling matrix $C$. The eigenvectors of $C$ are $\begin{pmatrix} 1 \\ 1 \end{pmatrix}$ and $\begin{pmatrix} 1 \\ -1 \end{pmatrix}$, with respective eigenvalues $-C_S + C_I$ and $-C_S - C_I$. We define $V$ to be the matrix whose columns are the eigenvectors: $V = \begin{pmatrix} 1 & 1 \\ 1 & -1 \end{pmatrix}$. Expressed in the eigenvector basis, the activity variables become the mean activity across bats, and the difference in activity between bats:

$$V^{-1} \begin{pmatrix} a_1(t) \\ a_2(t) \end{pmatrix} = \frac{1}{2} \begin{pmatrix} a_1(t) + a_2(t) \\ a_1(t) - a_2(t) \end{pmatrix} = \begin{pmatrix} a_M(t) \\ a_D(t) \end{pmatrix}.$$ Thus, the uncoupled activity variables are

precisely what we are interested in: the mean and difference components. Similarly, in the eigenvector basis, the behavioral modulation variables are the mean behavioral modulation and the difference in behavioral modulation: $V^{-1} \begin{pmatrix} b_1(t) \\ b_2(t) \end{pmatrix} = \frac{1}{2} \begin{pmatrix} b_1(t) + b_2(t) \\ b_1(t) - b_2(t) \end{pmatrix} = \begin{pmatrix} b_M(t) \\ b_D(t) \end{pmatrix}.$ Thus, in the eigenvector basis, *equation (2)* is

$$\tau \frac{d}{dt} \begin{pmatrix} a_M(t) \\ a_D(t) \end{pmatrix} = \begin{pmatrix} -C_S + C_I & 0 \\ 0 & -C_S - C_I \end{pmatrix} \begin{pmatrix} a_M(t) \\ a_D(t) \end{pmatrix} + \begin{pmatrix} b_M(t) \\ b_D(t) \end{pmatrix} \tag{3}$$

In the one-chamber model, $C_I > 0$, so functional across-brain coupling acts as positive feedback to the mean component and negative feedback to the difference component. The effects of such

opposite feedback to the two components can be seen more clearly by rewriting *equation (3)* as the following uncoupled differential equations:

$$\frac{\tau}{C_S - C_I} \frac{d\boldsymbol{a}_M(t)}{dt} = -\boldsymbol{a}_M(t) + \frac{\boldsymbol{b}_M(t)}{C_S - C_I} \tag{4}$$

$$\frac{\tau}{C_S + C_I} \frac{d\boldsymbol{a}_D(t)}{dt} = -\boldsymbol{a}_D(t) + \frac{\boldsymbol{b}_D(t)}{C_S + C_I} \tag{5}$$

Thus, at any given time $t$, $\boldsymbol{a}_M(t)$ is exponentially approaching $\frac{\boldsymbol{b}_M(t)}{C_S - C_I}$ with effective time constant $\frac{\tau}{C_S - C_I}$, and $\boldsymbol{a}_D(t)$ is exponentially approaching $\frac{\boldsymbol{b}_D(t)}{C_S + C_I}$ with effective time constant $\frac{\tau}{C_S + C_I}$. In the one-chamber model, $0 < C_I < C_S$, so $0 < C_S - C_I < C_S + C_I$. This means that in the one-chamber model, relative to the two-chambers model where $C_I = 0$, the positive feedback provided by functional across-brain coupling amplifies the mean component and slows it down. On the other hand, the negative feedback suppresses the difference component and speeds it up. Thus, the model suggests this opposite feedback to be a potential computational mechanism that explains all of our observations relating the magnitudes and timescales of the mean and difference components.

The only differences between the one-chamber and two-chambers models were the value of $C_I$ and the behavioral transition probability matrix. Having understood the effects of $C_I$, we now turn to the effects of the different transition probability matrices, which govern the Markov chain behavior models. For each Markov chain, we calculated its stationary distribution as the eigenvector of its transition probability matrix corresponding to an eigenvalue of 1, which was confirmed numerically in *Figure 3—figure supplement 2E, F*. *Figure 3—figure supplement 2E, F* shows that, for both the one-chamber and two-chambers models, the respective Markov chains approach stationary distributions in ~5 minutes, that is, the distribution of states no longer changes with time after ~5 min. The stationary distribution for the two-chambers model shows that states where the two bats engage in the same behavior are about equally likely as states where they engage in different behaviors (*Figure 3—figure supplement 2G*), as expected given that the two bats behave independently in separate chambers. For the one-chamber model, same-behavior states (probability 0.58) are more likely than different-behavior states (*Figure 3—figure supplement 2G*). The noiseless behavioral modulation $\boldsymbol{b}_d(t)$ depends on the behaviors of the two bats, while the behavioral modulation noise $\boldsymbol{b}_n(t)$ does not. Through $\boldsymbol{b}_d(t)$, same-behavior states contribute to behavioral modulation of the mean component, and the only noiseless behavioral modulation of the difference component comes from different-behavior states. In the one-chamber model, with increased coordinated behavioral modulation compared to the two-chambers model, the mean noiseless behavioral modulation across bats has larger magnitudes and slower timescales than its difference between bats (blue lines in *Figure 3—figure supplement 2H, I*). On the other hand, because the behavioral modulation noise is independent across bats, it does not distinguish between the mean and difference components. Thus, adding the noise to the noiseless behavioral modulation decreases the differential modulation of the mean and difference components (red lines in *Figure 3—figure supplement 2H, I*). However, even if the noise strongly reduces differential behavioral modulation of the mean and difference components, the experimentally observed levels of relative magnitudes and timescales can still result from functional across-brain coupling (purple lines in *Figure 3—figure supplement 2H, I*).

Thus, our model suggests two mechanisms behind our observations on the mean and difference components: opposite feedback by functional across-brain coupling, and coordinated behavioral modulation. Due to these mechanisms, in a shared social environment, the activity of the two bats become dominated by their common activity pattern, and when the activity of the two bats diverge, they rapidly converge again. As a result, the activity of two socially interacting bats are highly correlated with each other. We found that inter-brain correlation remained even when the bats were engaged in different behaviors (*Figure 4A–B*; see also *Zhang and Yartsev, 2019*), which suggests that coordinated behavioral modulation alone is not sufficient to explain the data, and that opposite feedback by functional across-brain coupling is needed. To explicitly test this, we performed an additional set of simulations of the one-chamber model. Here, after we generated each instantiation of $\boldsymbol{b}(t)$, we used it to simulate neural activity twice: once with functional across-brain coupling and once without. Note that the same $\boldsymbol{b}(t)$ is used for both simulations, including the same $\boldsymbol{b}_d(t)$ generated from the one-chamber Markov chain and the same instantiation of randomly generated $\boldsymbol{b}_n(t)$. *Figure 4C, D and G* shows that the model with functional across-brain coupling reproduced the experimental observation: inter-brain correlation remained after removing periods of coordinated behaviors. On the other hand,

without functional across-brain coupling, inter-brain correlation disappeared after removing coordinated behaviors (*Figure 4E–G*). Thus, we conclude that, in the framework of our model, opposite feedback by functional across-brain coupling is necessary to reproduce the data.

## Modeling: mean and difference components in a reduced model

To gain a more precise understanding how the relative variances and timescales of the mean and difference components depend on model parameters, we studied a reduced version of the model. In the reduced model, instead of generating $b(t)$ based on simulated behavior, $b_M(t)$ and $b_D(t)$ are simply noise with identical, flat power spectra. Such noise amounts to inputs with the same variance and same timescales to both the mean and difference components. Thus, the relative variances and timescales of the two components are solely determined by the coupling parameters $C_S$ and $C_I$. In the following, we derive expressions for the variance ratio and the power spectral centroid ratio of the mean and difference components, as functions of the coupling parameters.

Rewriting *equation (3)* and assuming $\tau = 1$ for simplicity (as $\tau$ is a redundant parameter and can be absorbed into the other parameters), the reduced model can be written as

$$\frac{da_M}{dt} = -(C_S - C_I) a_M + b_M$$
$$\frac{da_D}{dt} = -(C_S + C_I) a_D + b_D$$

The solutions to these equations are:

$$a_M(t) = a_M(0) f_M(t) + \int_0^t dt' b_M(t') f_M(t-t') \tag{6}$$

$$a_D(t) = a_D(0) f_D(t) + \int_0^t dt' b_D(t') f_D(t-t') \tag{7}$$

In the above, $f_M(t)$ and $f_D(t)$ are the neural decay functions for the mean and difference components:

$$f_M(t) = e^{-(C_S - C_I)t} = e^{\lambda_M t} \tag{8}$$

$$f_D(t) = e^{-(C_S + C_I)t} = e^{\lambda_D t} \tag{9}$$

where $\lambda_M = -C_S + C_I$ and $\lambda_D = -C_S - C_I$ are the eigenvalues of $C$.

The convolution theorem can be used to approximate the solutions (6) and (7) in the frequency domain:

$$\widetilde{a}_{M,k} \approx \sqrt{T} \widetilde{b}_{M,k} \widetilde{f}_{M,k} \tag{10}$$

$$\widetilde{a}_{D,k} \approx \sqrt{T} \widetilde{b}_{D,k} \widetilde{f}_{D,k} \tag{11}$$

Here, $T$ is the duration of a simulated session, and the tilde denotes a Fourier coefficient, for exmple, $\widetilde{f}_{M,k}$ is the Fourier coefficient of $f_M(t)$ at frequency $k$: $\widetilde{f}_{M,k} = \frac{1}{\sqrt{T}} \int_0^T dt\, f_M(t)\, e^{-i\frac{2\pi kt}{T}}$. The approximation of (10) and (11) assumes periodic convolutions and $a(0) = 0$ in (6) and (7), but it is still accurate when $a(0)$ is smaller or comparable to the scale of $b(t)$ and $T$ is large compared to the effective time constants $\frac{1}{C_S - C_I}$ and $\frac{1}{C_S + C_I}$.

We first consider the variance ratio. Using Parseval's theorem and (10), the variance of the mean component can be approximated in the frequency domain as

$$\begin{aligned} Var_M &= \frac{1}{T} \int_0^T dt \left[ a_M(t) - \frac{1}{T} \int_0^T dt' a_M(t') \right]^2 \\ &= \frac{2}{T} \sum_{k=1}^{\infty} \left| \widetilde{a}_{M,k} \right|^2 \\ &\approx 2 \sum_{k=1}^{\infty} \left| \widetilde{b}_{M,k} \right|^2 \left| \widetilde{f}_{M,k} \right|^2 \end{aligned}$$

In the reduced model, $\boldsymbol{b}_M$ has a flat power spectrum: $\left|\widetilde{\boldsymbol{b}}_{M,k}\right|^2 = p$ for all $k \geq 1$. Then, $Var_M \approx 2p \sum_{k=1}^{\infty} \left|\widetilde{f}_{M,k}\right|^2$. Transforming back to the time domain, plugging (8) in, and evaluating the integrals, we have

$$
\begin{aligned}
Var_M &\approx p \int_0^T dt\, f_M(t)^2 - \frac{p}{T}\left(\int_0^T dt\, f_M(t)\right)^2 \\
&= p\left(\frac{1}{2\lambda_M} - \frac{1}{T\lambda_M^2}\right)\left(e^{2\lambda_M T} - 1\right) + \frac{2p}{T\lambda_M^2}e^{\lambda_M T}
\end{aligned}
\tag{12}
$$

The same calculation applied to the variance of the difference component shows that it is

$$
Var_D \approx p\left(\frac{1}{2\lambda_D} - \frac{1}{T\lambda_D^2}\right)\left(e^{2\lambda_D T} - 1\right) + \frac{2p}{T\lambda_D^2}e^{\lambda_D T}
\tag{13}
$$

Assuming $\lambda_M < 0$ and $\lambda_D < 0$ (required for stability), for large $T$, (12) and (13) can be approximated as

$$
Var_M \approx -\frac{p}{2\lambda_M}
\tag{14}
$$

$$
Var_D \approx -\frac{p}{2\lambda_D}
\tag{15}
$$

Thus, the variance ratio (variance of the mean divided by variance of the difference) is

$$
r_{Var} = \frac{Var_M}{Var_D} \approx \frac{C_S + C_I}{C_S - C_I}
$$

Next, we consider the power spectral centroid ratio. Using (10), the power spectral centroid of the mean component can be approximated as

$$
\begin{aligned}
PSC_M &= \frac{\sum_{k=1}^{\infty} k\left|\widetilde{\boldsymbol{a}}_{M,k}\right|^2}{\sum_{j=1}^{\infty}\left|\widetilde{\boldsymbol{a}}_{M,j}\right|^2} \\
&\approx \frac{\sum_{k=1}^{\infty} k\left|\widetilde{\boldsymbol{b}}_{M,k}\right|^2\left|\widetilde{f}_{M,k}\right|^2}{\sum_{j=1}^{\infty}\left|\widetilde{\boldsymbol{b}}_{M,j}\right|^2\left|\widetilde{f}_{M,j}\right|^2} \\
&= \frac{\sum_{k=1}^{\infty} k\left|\widetilde{f}_{M,k}\right|^2}{\sum_{j=1}^{\infty}\left|\widetilde{f}_{M,j}\right|^2} \\
&= \frac{2p}{Var_M}\sum_{k=1}^{\infty} k\left|\widetilde{f}_{M,k}\right|^2
\end{aligned}
\tag{16}
$$

The Fourier coefficients of $f_M(t)$ are $\widetilde{f}_{M,k} = \frac{1}{\sqrt{T}}\int_0^T dt\, e^{\lambda_M t}e^{-i\frac{2\pi k t}{T}} = \frac{1}{\sqrt{T}}\frac{1}{\lambda_M - i\frac{2\pi k}{T}}\left(e^{\lambda_M T} - 1\right)$. Plugging this into (16), we have:

$$
PSC_M \approx \frac{2p}{Var_M}\left(e^{\lambda_M T} - 1\right)^2 \frac{T}{4\pi^2}\sum_{k=1}^{\infty}\frac{1}{\frac{\alpha_M}{k} + k}
$$

where $\alpha_M = \frac{T^2\lambda_M^2}{4\pi^2}$. The series $\sum_{k=1}^{\infty}\frac{1}{\frac{\alpha_M}{k}+k}$ can be shown to be divergent by comparison to $\frac{1}{\alpha_M+1}\sum_{k=1}^{\infty}\frac{1}{k}$, the harmonic series scaled by $\frac{1}{\alpha_M+1}$: $\frac{1}{\frac{\alpha_M}{k}+k} \geq \frac{1}{\alpha_M k+k}$ for all $k \geq 1$.

The same calculation applied to the power spectral centroid of the difference component shows that it is

$$
PSC_D \approx \frac{2p}{Var_D}\left(e^{\lambda_D T} - 1\right)^2 \frac{T}{4\pi^2}\sum_{k=1}^{\infty}\frac{1}{\frac{\alpha_D}{k}+k}
$$

where $\alpha_D = \frac{T^2\lambda_D^2}{4\pi^2}$.

While the power spectral centroids involve divergent series, we can define the power spectral centroid ratio (centroid of the mean divided by centroid of the difference) using a limit of the ratio of partial sums:

$$r_{PSC} \approx \frac{Var_D}{Var_M} \frac{\left(e^{\lambda_M T}-1\right)^2}{\left(e^{\lambda_D T}-1\right)^2} \lim_{n\to\infty} \frac{\sum_{k=1}^n \frac{1}{\frac{\alpha_M}{k}+k}}{\sum_{k=1}^n \frac{1}{\frac{\alpha_D}{k}+k}} \tag{17}$$

For simplicity, we rewrite the limit in (17) as $\lim_{n\to\infty} \frac{\beta_n^M}{\beta_n^D}$ where $\beta_n^M = \sum_{k=1}^n \frac{1}{\frac{\alpha_M}{k}+k}$ and $\beta_n^D = \sum_{k=1}^n \frac{1}{\frac{\alpha_D}{k}+k}$. Note that $\beta_n^D$ is strictly increasing and $\lim_{n\to\infty} \beta_n^D = \infty$, so we can evaluate the limit in (17) using the Stolz–Cesàro theorem:

$$\lim_{n\to\infty} \frac{\beta_n^M}{\beta_n^D} = \lim_{n\to\infty} \frac{\beta_{n+1}^M - \beta_n^M}{\beta_{n+1}^D - \beta_n^D} = \lim_{n\to\infty} \frac{\frac{\alpha_D}{n+1}+n+1}{\frac{\alpha_M}{n+1}+n+1} = 1$$

Thus, assuming $\lambda_M < 0$ and $\lambda_D < 0$, for large $T$, (17) can be approximated as

$$r_{PSC} \approx \frac{C_S - C_I}{C_S + C_I}$$

The results $r_{Var} \approx \frac{C_S + C_I}{C_S - C_I}$ and $r_{PSC} \approx \frac{C_S - C_I}{C_S + C_I}$ show the simple dependence of the variance ratio and power spectral centroid ratio on the coupling parameters (visualized in *Figure 3I–J*). The experimental results, where the mean component had higher variance and lower power spectral centroid than the difference component, correspond to the parameter regime of $0 < C_I < C_S$. In this regime, consistent with the earlier analysis of the full model, $C_I$ acts as positive feedback to the mean component to amplify it and slow it down, and acts negative feedback to the difference component to suppress it and speed it up.

## Modeling: Kuramoto model

We explored an alternative mechanism for inter-brain coupling using the Kuramoto model (*Strogatz, 2000*; *Acebrón et al., 2005*). In this model, the activity of the two brains are treated as two oscillators, whose phases are coupled:

$$\frac{d\theta_i(t)}{dt} = \omega_i + K \sum_{j=1}^2 sin\left(\theta_j(t) - \theta_i(t)\right), \; i = 1, 2 \tag{18}$$

Here $\theta_1(t)$ and $\theta_2(t)$ are the phases of the two oscillators at time $t$, and the corresponding activity of the two brains at time $t$ are $a_1(t) = \frac{sin(\theta_1(t))+1}{2}$ and $a_2(t) = \frac{sin(\theta_2(t))+1}{2}$; $\omega_1$ and $\omega_2$ are the natural frequencies of the two oscillators; $K$ is the coupling strength between the oscillators: $K > 0$ for simulations of one-chamber sessions, and $K = 0$ for simulations of two-chambers sessions.

For each simulation, $\omega_1$ and $\omega_2$ were drawn from lognormal distributions with means $\bar{\omega}_1$ and $\bar{\omega}_2$, respectively, and the same standard deviation $\sigma_\omega$. $\bar{\omega}_1$, $\bar{\omega}_2$, and $\sigma_\omega$ were set to make $\omega_1$ and $\omega_2$ different from each other, to avoid the two brains trivially synchronizing by having the same natural frequency. To simulate the Kuramoto model, *equation (18)* was numerically integrated (*ode15s* function in MATLAB) with initial condition $\theta_1(0) = \theta_2(0) = 0$. The phases $\theta_1(t)$ and $\theta_2(t)$ were then converted to activity $a_1(t)$ and $a_2(t)$, which were plotted in *Figure 4—figure supplement 1A-D* and analyzed in *Figure 4—figure supplement 1E-F*. This shows that the phase-coupling mechanism of the Kuramoto model is able to reproduce inter-brain correlation and the relative magnitudes of the mean and difference components from the data; however, it does not reproduce the relative timescales of the mean and difference components from the data.

## Modeling: correlation between activity variables under rotated activity bases

The neural dynamics in the model are governed by (2). To express (2) under a different basis, $a(t)$, $C$, and $b(t)$ are transformed to $a'(t) = U^{-1} a(t)$, $C' = U^{-1} C U$, and $b'(t) = U^{-1} b(t)$, respectively, where the columns of the matrix $U$ are the new basis vectors. If $U = \begin{pmatrix} \cos\theta & -\sin\theta \\ \sin\theta & \cos\theta \end{pmatrix}$, then the new basis corresponds to a counter-clockwise rotation of the original basis (where each axis is the activity of one bat) by an angle of $\theta$. By rotating the basis, the functional coupling between the activity variables (i.e. between the two elements of $a'(t)$) changes as a function of the rotation angle (*Figure 5A and D*). This functional coupling in turn determines the correlations between those activity variables, which

forms a set of predictions of the model (*Figure 5B and E*). These predictions are confirmed in the data in *Figure 5C and F*.

Additionally, we examined the influence of behavioral modulation on correlation between the activity variables under rotated bases, by regressing out behavior from the activity variables. To do this, for each behavior (e.g. resting, self-grooming, probing, etc.), each bat, and each simulation or experimental session, we used a binary vector to represent the time course of the bat engaging in that behavior: the elements of the vector correspond to time points, and the values of the elements are '1' at time points when the bat was engaging in that behavior, and '0' when it was not. Then, we performed linear regression to predict each activity variable over time under each basis using the set of all behavioral binary vectors of both bats as predictors. The linear regression fit was then subtracted from the activity, and correlation was calculated between these residuals. The results are shown in *Figure 5B, C, E and F* (brown lines): for both model and data, regressing out behaviors changes the magnitudes, but not the shapes of the correlation curves.

## Modeling: relationship between variance ratio and power spectral centroid ratio

Here we analyze the relationship between the mean/difference ratio of the variance ($r_{Var}$) and the mean/difference ratio of the power spectral centroid ($r_{PSC}$) in the model (seen in *Figure 5G*). $r_{Var}$ and $r_{PSC}$ are functions of the behavioral modulation $\boldsymbol{b}(t)$, which differs from simulation to simulation. *Figure 5G* shows that $r_{Var}$ and $r_{PSC}$ corresponding to different instances of $\boldsymbol{b}(t)$ tend to covary linearly. To examine whether this linear relationship is due to a systematic relationship between different instances of $\boldsymbol{b}(t)$, or due to the neural dynamics, we examine the effect of random perturbations to behavioral modulation on the relationship between $r_{Var}$ and $r_{PSC}$. To do so, we first express $r_{Var}$ and $r_{PSC}$ in the frequency domain, and then examine whether and how they covary given random perturbations.

We first follow the procedure of the earlier section on the reduced model to approximate the variance and the power spectral centroid. In that section, we analyzed the reduced model with $\tau = 1$ and $\left|\tilde{\boldsymbol{b}}_{M,k}\right|^2 = \left|\tilde{\boldsymbol{b}}_{D,k}\right|^2 = p$ for $k \geq 1$; here we consider arbitrary $\tau$, $\left|\tilde{\boldsymbol{b}}_{M,k}\right|^2$, and $\left|\tilde{\boldsymbol{b}}_{D,k}\right|^2$. In this case, the neural decay functions are $f_M(t) = e^{-\frac{c_S - c_I}{\tau}t}$ and $f_D(t) = e^{-\frac{c_S + c_I}{\tau}t}$, and the variance and power spectral centroid of the mean component are

$$Var_M \approx \frac{2}{\tau^2} \sum_{k=1}^{\infty} \boldsymbol{B}_{M,k} F_{M,k}$$

$$PSC_M \approx \frac{\sum_{k=1}^{\infty} k \boldsymbol{B}_{M,k} F_{M,k}}{\sum_{j=1}^{\infty} \boldsymbol{B}_{M,j} F_{M,j}}$$

where $\boldsymbol{B}_{M,k} = \left|\tilde{\boldsymbol{b}}_{M,k}\right|^2$ and $F_{M,k} = \left|\tilde{f}_{M,k}\right|^2$. The expressions for the difference component are similar.

Then, the mean/difference ratio of the variance is

$$r_{Var} \approx \frac{\sum_{k=1}^{\infty} \boldsymbol{B}_{M,k} F_{M,k}}{\sum_{j=1}^{\infty} \boldsymbol{B}_{D,j} F_{D,j}} \tag{19}$$

The mean/difference ratio of the power spectral centroid is

$$r_{PSC} \approx \frac{\sum_{k=1}^{\infty} k \boldsymbol{B}_{M,k} F_{M,k}}{\sum_{j=1}^{\infty} \boldsymbol{B}_{M,j} F_{M,j}} \frac{\sum_{h=1}^{\infty} \boldsymbol{B}_{D,h} F_{D,h}}{\sum_{g=1}^{\infty} g \boldsymbol{B}_{D,g} F_{D,g}} \tag{20}$$

To examine how $r_{Var}$ and $r_{PSC}$ change with changes in behavioral modulation, we calculate the following partial derivatives. The partial derivative of $r_{Var}$ with respect to the power of the mean component of behavioral modulation at frequency $k$ is

$$\frac{\partial r_{Var}}{\partial \boldsymbol{B}_{M,k}} = \frac{1}{\sum_{j=1}^{\infty} \boldsymbol{B}_{D,j} F_{D,j}} F_{M,k} \tag{21}$$

The partial derivative of $r_{Var}$ with respect to the power of the difference component of behavioral modulation at frequency $k$ is

$$\frac{\partial r_{Var}}{\partial \boldsymbol{B}_{D,k}} = -\frac{\sum_{h=1}^{\infty} \boldsymbol{B}_{M,h} F_{M,h}}{\left(\sum_{j=1}^{\infty} \boldsymbol{B}_{D,j} F_{D,j}\right)^2} F_{D,k} \tag{22}$$

The partial derivative of $r_{PSC}$ with respect to the power of the mean component of behavioral modulation at frequency $k$ is

$$\frac{\partial r_{PSC}}{\partial \boldsymbol{B}_{M,k}} = \frac{\sum_{j=1}^{\infty} \boldsymbol{B}_{D,j} F_{D,j}}{\left(\sum_{h=1}^{\infty} \boldsymbol{B}_{M,h} F_{M,h}\right) \left(\sum_{q=1}^{\infty} q \boldsymbol{B}_{D,q} F_{D,q}\right)} F_{M,k} \left(k - \frac{\sum_{p=1}^{\infty} p \boldsymbol{B}_{M,p} F_{M,p}}{\sum_{g=1}^{\infty} \boldsymbol{B}_{M,g} F_{M,g}}\right) \tag{23}$$

The partial derivative of $r_{PSC}$ with respect to the power of the difference component of behavioral modulation at frequency $k$ is

$$\frac{\partial r_{PSC}}{\partial \boldsymbol{B}_{D,k}} = \frac{\sum_{p=1}^{\infty} p \boldsymbol{B}_{M,p} F_{M,p}}{\left(\sum_{g=1}^{\infty} \boldsymbol{B}_{M,g} F_{M,g}\right) \left(\sum_{l=1}^{\infty} l \boldsymbol{B}_{D,l} F_{D,l}\right)} F_{D,k} \left(1 - k \frac{\sum_{j=1}^{\infty} \boldsymbol{B}_{D,j} F_{D,j}}{\sum_{q=1}^{\infty} q \boldsymbol{B}_{D,q} F_{D,q}}\right) \tag{24}$$

In subsequent calculations, the infinite sums were truncated at $k_t$ such that $\frac{k_t}{T} = 0.2$ Hz, which is the Nyquist frequency for the sampled simulated activity used for our analyses.

We now consider perturbations to the power spectra of behavioral modulation. We concatenate the power spectra of the behavioral modulation of the mean and difference components to form

$$\boldsymbol{B} = \begin{pmatrix} \boldsymbol{B}_{M,1} \\ \vdots \\ \boldsymbol{B}_{M,k_t} \\ \boldsymbol{B}_{D,1} \\ \vdots \\ \boldsymbol{B}_{D,k_t} \end{pmatrix}$$

We perturb a given $\boldsymbol{B}$ by adding $\delta\boldsymbol{B}$, drawn from a uniform distribution on the hypersphere centered at 0. To see whether and how $r_{Var}$ and $r_{PSC}$ covary given random perturbations to behavioral modulation, we next determine the covariance matrix between the changes in $r_{Var}$ and $r_{PSC}$ resulting from perturbations $\delta\boldsymbol{B}$.

Considering $r_{Var}$ and $r_{PSC}$ as functions of $\boldsymbol{B}$, their gradients are

$$\nabla r_{Var} = \begin{pmatrix} \frac{\partial r_{Var}}{\partial \boldsymbol{B}_{M,1}} \\ \vdots \\ \frac{\partial r_{Var}}{\partial \boldsymbol{B}_{M,k_t}} \\ \frac{\partial r_{Var}}{\partial \boldsymbol{B}_{D,1}} \\ \vdots \\ \frac{\partial r_{Var}}{\partial \boldsymbol{B}_{D,k_t}} \end{pmatrix}, \nabla r_{PSC} = \begin{pmatrix} \frac{\partial r_{PSC}}{\partial \boldsymbol{B}_{M,1}} \\ \vdots \\ \frac{\partial r_{PSC}}{\partial \boldsymbol{B}_{M,k_t}} \\ \frac{\partial r_{PSC}}{\partial \boldsymbol{B}_{D,1}} \\ \vdots \\ \frac{\partial r_{PSC}}{\partial \boldsymbol{B}_{D,k_t}} \end{pmatrix}$$

where the partial derivatives are given in (21)-(24). Given a small perturbation $\delta\boldsymbol{B}$, the resulting changes to $r_{Var}$ and $r_{PSC}$ are approximately $\delta\boldsymbol{B}^T \nabla r_{Var}$ and $\delta\boldsymbol{B}^T \nabla r_{PSC}$, respectively. To calculate the covariance matrix between $\delta\boldsymbol{B}^T \nabla r_{Var}$ and $\delta\boldsymbol{B}^T \nabla r_{PSC}$, we first perform an orthogonal transformation to any orthonormal basis whose first basis vector is $\nabla r_{Var}/\|\nabla r_{Var}\|$, where $\|\cdot\|$ denotes vector norm. We use $\delta\hat{\boldsymbol{B}}$, $\nabla \hat{r}_{Var}$, and $\nabla \hat{r}_{PSC}$ to denote $\delta\boldsymbol{B}$, $\nabla r_{Var}$, and $\nabla r_{PSC}$ under the new basis, respectively. The variance of $\delta\boldsymbol{B}^T \nabla r_{Var}$ is

$$\begin{aligned} Var\left(\delta\boldsymbol{B}^T \nabla r_{Var}\right) &= Var\left(\delta\hat{\boldsymbol{B}}^T \nabla \hat{r}_{Var}\right) \\ &= E\left(\left(\delta\hat{\boldsymbol{B}}^T \nabla \hat{r}_{Var}\right)^2\right) \\ &= \|\nabla r_{Var}\|^2 E\left(\delta\hat{\boldsymbol{B}}_1^2\right) \\ &= \frac{\|\delta\boldsymbol{B}\|^2}{2k_t} \|\nabla r_{Var}\|^2 \end{aligned}$$

Similarly, the variance of $\delta \boldsymbol{B}^T \nabla r_{PSC}$ is $\frac{\|\delta \boldsymbol{B}\|^2}{2k_t} \|\nabla r_{PSC}\|^2$. The covariance between $\delta \boldsymbol{B}^T \nabla r_{Var}$ and $\delta \boldsymbol{B}^T \nabla r_{PSC}$ is

$$
\begin{aligned}
cov\left(\delta \boldsymbol{B}^T \nabla r_{Var}, \delta \boldsymbol{B}^T \nabla r_{PSC}\right) &= cov\left(\delta \hat{\boldsymbol{B}}^T \nabla \hat{r}_{Var}, \delta \hat{\boldsymbol{B}}^T \nabla \hat{r}_{PSC}\right) \\
&= E\left(\delta \hat{\boldsymbol{B}}^T \nabla \hat{r}_{Var}\, \delta \hat{\boldsymbol{B}}^T \nabla \hat{r}_{PSC}\right) \\
&= E\left(\delta \hat{\boldsymbol{B}}_1 \|\nabla r_{Var}\| \sum_{j=1}^{2k_t} \delta \hat{\boldsymbol{B}}_j \nabla \hat{r}_{PSCj}\right) \\
&= E\left(\delta \hat{\boldsymbol{B}}_1^2 \|\nabla r_{Var}\| \frac{\nabla r_{Var}^T}{\|\nabla r_{Var}\|} \nabla r_{PSC} + \sum_{j=2}^{2k_t} \delta \hat{\boldsymbol{B}}_1 \delta \hat{\boldsymbol{B}}_j \|\nabla r_{Var}\| \nabla \hat{r}_{PSCj}\right) \\
&= \nabla r_{Var}^T \nabla r_{PSC} E(\delta \hat{\boldsymbol{B}}_1^2) \\
&= \frac{\|\delta \boldsymbol{B}\|^2}{2k_t} \nabla r_{Var}^T \nabla r_{PSC}
\end{aligned}
$$

Thus, the covariance matrix between $\delta \boldsymbol{B}^T \nabla r_{Var}$ and $\delta \boldsymbol{B}^T \nabla r_{PSC}$ is

$$
\frac{\|\delta \boldsymbol{B}\|^2}{2k_t} \begin{pmatrix} \|\nabla r_{Var}\|^2 & \nabla r_{Var}^T \nabla r_{PSC} \\ \nabla r_{Var}^T \nabla r_{PSC} & \|\nabla r_{PSC}\|^2 \end{pmatrix} \tag{25}
$$

The eigenvector of this matrix corresponding to the larger eigenvalue is the direction of the local linear trend, whereas the relative sizes of the eigenvalues indicate the strength of the linear trend (the larger the difference between the eigenvalues, the stronger the linear trend).

We examined the eigenvectors and eigenvalues of (25) across different simulations (*Figure 5—figure supplement 1*; in computing (25), integrals were evaluated numerically). We found that $r_{Var}$ and $r_{PSC}$ resulting from random perturbations to $\boldsymbol{B}$ consistently show linear relationships. Furthermore, the slopes of these local linear relationships, which are not influenced by systematic variations of behavioral modulation $\boldsymbol{b}(t)$ across simulations, are consistent with the slopes of the global linear relationships seen in *Figure 5G*.

## Modeling: using the neural activity of one bat to discriminate the behavior of the other bat

In the model, the neural activity of each bat is directly modulated by its own behavior (e.g. $\boldsymbol{a}_1(t)$ is modulated by $\boldsymbol{b}_1(t)$). Additionally, in the one-chamber model, the activity of each bat is indirectly modulated by the behavior of the other bat, through functional across-brain coupling (e.g. $\boldsymbol{a}_1(t)$ is modulated by $\boldsymbol{b}_2(t)$ through the coupling $\boldsymbol{C}_I$). This naturally suggests that the neural activity of each bat should encode the behavior of the other bat independently from encoding its own behavior.

We used the following method to quantify this. Before examining whether the activity of one bat can discriminate the behavior of the other bat, we first regressed out the behavior of each bat from its own neural activity, using the method described in the earlier section 'Modeling: correlation between activity variables under rotated activity bases'. Importantly, this eliminates potential spurious correlation between one bat's activity and another bat's behavior caused by any coordinated behaviors between the bats. We then asked whether the neural activity of a given bat discriminates between a given pair of behaviors by the other bat (e.g., using bat 1's neural activity to discriminate whether bat 2 is resting or engaging in social grooming), by plotting the receiver operating characteristic (ROC) curve and calculating the area under the curve (*Figure 6B and E*; *Dayan and Abbott, 2005*). For each discrimination, positive and negative class assignments for the two behaviors were made so that the area under the ROC curve is greater than or equal to 0.5. We then averaged the area under the ROC curve across bats, pairs of behaviors, and simulations or sessions (*Figure 6C and F*). This showed that the activity of each bat was modulated by the behaviors of the other bat independently of its own behavior, in both the model and the data.

We note that for both the model and the data, the area under the ROC curve was above 0.5 for the two-chambers sessions—this is a necessary consequence of noise and finite sample size. To illustrate this, we can consider a hypothetical example where neural activity does not encode behaviors A and B, that is, the distributions of neural activity during these two behaviors are identical. If we

have infinite amount of data, or if the neural activity is noiseless during these two behaviors, then the empirically observed distributions of activity during these two behaviors would be exactly identical, so that the area under the ROC curve would be exactly 0.5. However, in reality we have finite amounts of noisy data, so that the two empirically observed distributions would not be identical. These different empirical distributions would then necessarily result in an area under the ROC curve greater than 0.5, since positive and negative classes were assigned to the two behaviors such that the area under the ROC curve is greater than or equal to 0.5.

## Modeling: inter-brain relationship during group social interactions

To generalize our two-bat model to more than two bats, we used the same *equation (2)*, with $a(t)$ and $b(t)$ now being $n$-dimensional vectors, and $C$ being an $n \times n$ matrix, where $n$ is the number of interacting bats. $C$ retains the same structure from the two-bat model: all diagonal elements (functional self-coupling) are $-C_S$, and all off-diagonal elements (functional across-brain coupling) are $C_I$.

To understand the $n$-bat model, we examine the eigenvectors and eigenvalues of $C$. Note that the $n$-dimensional vector whose elements are all 1s is an eigenvector, with eigenvalue $(n-1)C_I - C_S$. This eigenvector corresponds to the direction of the mean activity across all bats, and is thus the $n$-bat analogue of the mean component from the two-bat model. Any vector orthogonal to the $n$-bat mean component is also an eigenvector, with eigenvalue $-C_I - C_S$. These eigenvectors define an $(n-1)$-dimensional subspace, which contains all inter-brain activity patterns that correspond to activity differences across brains; we call this subspace the difference subspace, which is the multi-dimensional analogue of the difference component from the two-bat model. To ensure stability (so that neural activity do not go to infinity), we take our parameter regime to be $0 < (n-1)C_I < C_S$. Thus, $-C_I - C_S < (n-1)C_I - C_S < 0$. This means that, similar to the two-bat model, the $n$-bat mean component is amplified and slowed down by the positive feedback provided by functional across-brain coupling, whereas activity patterns in the difference subspace are suppressed and sped up by negative feedback. This results in the predictions that, for $n$ socially interacting bats, activity in the $n$-bat mean component will have larger magnitude and slower timescales than activity in the difference subspace on average (*Figure 7C–D*).

Another prediction concerns the correlation between activity variables during group interactions. Because the activity of pairs of bats are positively functionally coupled in the $n$-bat model, the model would predict positive inter-brain correlations (*Figure 7E*). On the other hand, because the $n$-bat mean component and vectors in the difference subspace are all eigenvectors, they are functionally uncoupled. Thus, the model would predict zero average correlation between the $n$-bat mean component and activity variables in the difference subspace (*Figure 7E*).

For the simulations in *Figure 7C–E*, we used an $n$-bat model with $n = 4$ and performed the simulations using the same procedures as for the two-bat model. Because we do not have behavioral statistics for four-bat group interactions, we opted to use noise for $b(t)$, so that model predictions based on functional across-brain coupling would not be biased by structures in $b(t)$ (e.g. the $n$-bat mean component could have larger magnitudes than the difference subspace if $b(t)$ has such a structure). To generate $b(t)$ for the $n$-bat model, we used the following procedure. For a given simulated session and a given bat $i$, we generated $b_i(t)$ independently of the other bats. We generated a random vector $b_{pre}$, whose dimensionality was the number of time points spaced 2.5 s apart in the simulated session (matching the time step size of the Markov chain from the two-bat model). The elements of $b_{pre}$ were drawn independently from a Gaussian distribution with mean $b_{mean}$ and standard deviation $b_{std}$. We smoothed $b_{pre}$ with a 1200-point moving average filter, then added independent Gaussian noise to its elements (0 mean, standard deviation $\sigma_n$) as in the two-bat model. The resulting vector contained the values of $b_i(t)$ at 2.5 s intervals; $b_i(t)$ at time points in-between were linearly interpolated. The same procedure was repeated for each bat. Note that, because $b(t)$ was randomly generated noise, the simulations only offered qualitative predictions. Indeed, the data confirmed the qualitative trends seen in the simulations, but differed from them quantitatively (*Figure 7C–H*).

## Modeling: parameters

In our data, all four neural signals (30–150 Hz LFP power, 1–29 Hz LFP power, multiunits, and single units) show the same qualitative phenomena: faster and smaller difference component compared to

the mean component, on one-chamber sessions but not two-chambers sessions (*Figure 1*). The four signals differed quantitatively in the extent they showed the same qualitative phenomena. The goal of our model is not to quantitatively fit any one of the four neural signals in particular, but to provide mechanistic explanations for the general qualitative phenomena that do not require fine-tuning of parameters. As explained above, the opposite feedback mechanism depends simply on $0 < C_I < C_S$, whereas the coordinated behavioral modulation mechanism is a manifestation of the empirical behavioral transition frequencies.

The focus of the model is to reproduce and explain the qualitative trend of the difference component being faster and smaller than the mean component. Other aspects of the data could be reproduced by extending the model with additional parameters, but we chose not to do so in the interest of keeping the model simple and focused. For example, the 30–150 Hz LFP power data showed more variability across sessions compared to the model simulations (*Figure 3E–F*). The higher variability could be reproduced if we introduced variability across simulations in the behavioral transition probabilities or the strength of functional across-brain coupling.

The model parameters for the two-bat models are: $C_S = 1$, $C_I = 0.4$ for simulations with functional across-brain coupling, $C_I = 0$ for simulations without functional across-brain coupling, $\tau = 15s$, $\sigma_n = 0.15$, $T = 100$ minutes, $b_{resting} = 0.158$, $b_{active\,non-social} = 0.269$, $b_{self-grooming} = 0.264$, $b_{social\,grooming} = 0.223$, $b_{probing} = 0.284$, $b_{fighting} = 0.355$, $b_{mating} = 0.367$, $b_{wing\,covering} = 0.364$, $b_{reaching} = 0.321$, $b_{blocking} = 0.339$, $b_{other\,interactions} = 0.291$, $b_{constant} = -0.08$ for one-chamber simulations and $b_{constant} = 0$ for two-chambers simulations. The behavioral modulation parameter for each behavior listed above ($b_{resting}$, etc.) was set as the average 30–150 Hz mean normalized LFP power during that behavior from the data: take a given bat, for each session, average its 30–150 Hz mean normalized LFP power across all channels, pool together these averaged power values from all time points when the bat engaged in the given behavior from all sessions (including both one-chamber and two-chambers), then pool across all bats, and then average across all such pooled data. $b_{constant}$, $C_S$, $C_I$, $\tau$, and $\sigma_n$ were chosen so that the levels of simulated neural activity roughly match the levels of 30–150 Hz mean normalized LFP power observed during the experiments.

The model parameters for the four-bat model are: $C_S = 1$, $C_I = 0.1$, $b_{mean} = 0.2$, and $b_{std} = 3.5$. All other parameters are the same as in the two-bat model.

The model parameters for the Kuramoto model are: $K = 0.0035$ for simulations of one-chamber sessions, $K = 0$ for simulations of two-chambers sessions, $\bar{\omega}_1 = 0.005$, $\bar{\omega}_2 = 0.01$, and $\sigma_\omega = 0.0005$.

## Sample sizes

In this section we list the sample sizes for all results that were presented as averages.

*Figure 1M, N, Q and R*: n = 52 sessions for one-chamber sessions, and 18 sessions for two-chambers sessions.

*Figure 1O and S*: n = 675 multiunit pairs for one-chamber sessions, and 284 multiunit pairs for two-chambers sessions.

*Figure 1P and T*: n = 256 single unit pairs for one-chamber sessions, and 65 single unit pairs for two-chambers sessions.

*Figure 3E–F*: for data, n = 52 sessions for one-chamber sessions, and 18 sessions for two-chambers sessions; for model, n = 100 simulations for one-chamber model, and 100 simulations for two-chambers model.

*Figure 4G*: for data, n = 50 sessions; for model, n = 100 simulations with functional across-brain coupling, and 100 simulations without functional across-brain coupling.

*Figure 5B*: n = 100 simulations.

*Figure 5C*: n = 52 sessions.

*Figure 5E*: n = 100 simulations.

*Figure 5F*: n = 18 sessions.

*Figure 6C*: n = 8,235 simulations × bats × behavior pairs for one-chamber simulations, n = 1,848 simulations × bats × behavior pairs for two-chambers simulations.

*Figure 6F*: n = 2086 sessions × bats × behavior pairs for one-chamber sessions, n = 200 sessions × bats × behavior pairs for two-chambers sessions.

*Figure 7C–E*: n = 100 simulations.

*Figure 7F–H*: n = 20 sessions.

*Figure 3—figure supplement 2A, B*: n = 100 cross-validation test sets.

*Figure 3—figure supplement 2H, I*: for data, n = 52 sessions for one-chamber sessions, and 18 sessions for two-chambers sessions; for model, n = 100 simulations for one-chamber model, and 100 simulations for two-chambers model.

*Figure 4—figure supplement 1E, F*: for data, n = 52 sessions for one-chamber sessions, and 18 sessions for two-chambers sessions; for the Kuramoto model, n = 100 simulations for one-chamber model, and 100 simulations for two-chambers model.

*Figure 5—figure supplement 1C, E*: n = 100 simulations for one-chamber model, and 100 simulations for two-chambers model.

## Acknowledgements

We thank ES Sevilla, A Raha, J Chau, K Moi, N Juthani, M Zuercher, L Kasraie, and C Tran for behavioral annotation; S A Afjei for histology; A Halley and L Krubitzer for providing the image of the 3D-reconstructed brain; L Jiang for machining; B Olshausen, W Liberti, and members of the Yartsev Lab for discussion and comments on the manuscript; C Ferrecchia and G Lawson for veterinary oversight; the staff of the Office of Laboratory Animal Care for support with animal husbandry and care. This research was supported by NIH (DP2-DC016163), NIMH (1-R01MH25387-01), the New York Stem Cell Foundation (NYSCF-R-NI40), the Alfred P Sloan Foundation (FG-2017–9646), the Brain Research Foundation (BRFSG-2017–09), National Science Foundation (NSF- 1550818), the Packard Fellowship (2017–66825), the Klingenstein-Simons Fellowship, the Pew Charitable Trust (00029645), and the Dana Foundation (to MMY).

## Additional information

### Funding

| Funder | Grant reference number | Author |
|---|---|---|
| National Institutes of Health | DP2-DC016163 | Michael M Yartsev |
| National Institute of Mental Health | 1-R01MH25387-01 | Michael M Yartsev |
| New York Stem Cell Foundation | NYSCF-R-NI40 | Michael M Yartsev |
| Alfred P. Sloan Foundation | FG-2017-9646 | Michael M Yartsev |
| Brain Research Foundation | BRFSG-2017-09 | Michael M Yartsev |
| National Science Foundation | NSF- 1550818 | Michael M Yartsev |
| David and Lucile Packard Foundation | 2017-66825 | Michael M Yartsev |
| Simons Foundation | | Michael M Yartsev |
| Pew Charitable Trusts | 00029645 | Michael M Yartsev |
| Dana Foundation | | Michael M Yartsev |

The funders had no role in study design, data collection and interpretation, or the decision to submit the work for publication.

### Author contributions

Wujie Zhang, Conceptualization, Data curation, Formal analysis, Investigation, Methodology, Software, Validation, Visualization, WZ collected the two-bat data and performed the analysis and modeling, with help and feedback from MMY, WZ and MMY wrote the manuscript with input from MCR, Writing – original draft, Writing – review and editing; Maimon C Rose, Data curation, Investigation, MCR

collected the four-bat data, with help and feedback from MMY, Writing – review and editing; Michael M Yartsev, Conceptualization, Funding acquisition, Investigation, MMY supervised the entire study, WZ and MMY wrote the manuscript with input from MCR, Methodology, Supervision, Writing – review and editing

**Author ORCIDs**
Wujie Zhang ORCID http://orcid.org/0000-0002-1019-8247
Michael M Yartsev ORCID http://orcid.org/0000-0003-0952-2801

**Ethics**
All experimental procedures complied with all relevant ethical regulations for animal testing and research and were approved by the Institutional Animal Care and Use Committee of the University of California, Berkeley (protocol number AUP-2015-01-7122-2).

**Decision letter and Author response**
Decision letter https://doi.org/10.7554/eLife.70493.sa1
Author response https://doi.org/10.7554/eLife.70493.sa2

# Additional files

**Supplementary files**
• Transparent reporting form

**Data availability**
Source code for the models is available at https://github.com/zhang-wujie/Neurobat-lab-codes/tree/master/Interbrain-model, copy archived at swh:1:rev:20f445028b5b1fa3b8b15be020252a397eb6479f.

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
