## [Decision Letter]

**Decision letter after peer review:**

Thank you for submitting your article "A unifying mechanism governing inter-brain neural relationship during social interactions" for consideration by *eLife*. Your article has been reviewed by 3 peer reviewers, including Noah Cowan as the Reviewing Editor and Reviewer #1, and the evaluation has been overseen by Ronald Calabrese as the Senior Editor. Two of the reviewers have opted to remain anonymous.

The reviewers have discussed their reviews with one another, and the Reviewing Editor has drafted this decision letter.

After careful consideration and discussion, the reviewers concur that there is some potential for this work, but the reviewers are unanimous in noting that the paper has serious weaknesses and requires extensive new analysis and a major revision to be suitable for publication in *eLife*. We note that this in part stems from the high bar of publication in *eLife*; this paper may be suitable in a more field specific journal as is, and so the authors will need to balance the requirements we are placing for new revisions and the desire to publish the work as is.

Also, it was noted that this manuscript reuses portions of the methods section of a prior article by the same authors (Zhang & Yartsev 2019 Cell). The methods should be rewritten so as to avoid this and it should be made clear when you are quoting directly from the prior manuscript.

If the authors choose to revise the manuscript for *eLife*, it is not guaranteed that future revisions will be accepted, as the model would still need to be vetted by the reviewers.

*Reviewer #1:*

This paper provides experimental and modeling analysis of the inter-brain coupling of socially interacting bats, and reports that coordinated brain activity evolves at a slower time scale than the activity describing the differences. Specifically, the paper finds that there is an attracting submanifold corresponding to the mean (or "common mode") of neural activity, and that the dynamics in the orthogonal eigenmode, corresponding to the difference in brain activity, decays rapidly. These rapid decays in the difference mode are referred to as "catch up" activity.

There are two main findings:

1) Neural activity (especially higher frequency LFP activity in the 30-150Hz range) is modulated by social context. Specifically, the ratio of the averaged, moment-to-moment MEAN:DIFF ratio is much higher when the bats are in a single chamber, clearly indicating that the animals are coordinating their neural activity. This change also seems to hold -- although not as striking -- in lower-frequency LFP and spiking activity.

2) The time scales of the mean vs. difference dynamics are segregated: the "difference dynamics" evolve at a faster time scale than "similarity dynamics", seems to be well supported.

The basic finding is presented in Figure 1. The rest of the paper is focused on a modeling study to garner further insight into the dynamics.

Weaknesses:

This is an entirely phenomenological paper, and while it claims to garner "mechanistic insight", it is unclear what that means.

The basic idea of the model is simple and somewhat interesting, but the details are extremely complex. There are many examples of this, but the method used to "regress out" the behavior was very hard to interpret.

On the face of it, the model is extremely simple: a two-state linear dynamical system. However, this simplistic description buries extreme complexity. The model is extremely complex as involves a large number of parameters (e.g., time switching 'b' values, the values of which are completely unclear), the switching over time of these parameters based on hand-scored animal behavioral state, and the complex mix of markovian and linear dynamical systems theoretic results. Indeed, a fundamental weakness of the model is that the Markov chain is taken as an "input" to the 2-state linear systems model, as if somehow the neural state does not affect the state transitions. Further, the Markov assumption is not rigorously tested. No model selecting or other model validation appears to be done.

In short, the model, while very interesting, is so complex that it is literally impossible to evaluate. The authors report literally no shortcomings of their model. They do not report parameter estimation methods. They do not report fitting errors or other model validation metrics. The only evaluation is whether it can produce certain outputs that are similar to biological data. While the latter is certainly important, all models are wrong, and it essential to have a model simple enough to understand, both in terms of how it works and how it fails.

In general, while the basic finding is fairly interesting, and the experiments and their findings are highly relevant to the field, the modeling and its explication fall short.

It is not that it is wrong or bad; however, it is not clear that such a complex model increases our understanding beyond the experimental findings in Figure 1, and if it does, there has to be a major caveat that the model itself is not carefully vetted.

*Reviewer #2:*

In this paper, Wujie Zhang and Michael Yartsev investigate some of the basic underpinnings of inter-brain synchrony in socially interacting animals. The phenomena of inter brain synchrony is fascinating and has been observed in a variety of situations across different mammalian species. It has also been proposed to play a critical role in certain social behaviors. Here, the authors report that brain activity across two interacting bats display not only similarities but also important differences. The also use advance computational modeling to capture s these two characteristics as well as to demonstrate how they are affected by the presence and absence of interaction between animal pairs.

*Reviewer #3:*

The activity in the frontal cortex of mammals has been previously shown to become more correlated in socially interacting animals than when they are alone. In the current study, the authors examine the differences in brain activity that emerge during social interactions. The correlations and differences in activity were shown to occur over different time scales, with mean correlations occurring over longer time scales whereas differences occur over shorter time scales. The authors made a model of these processes that show how feedback may give rise to these phenomena.

---

## [Author Response]

Essential revisions:1) Extensive rewrite of the manuscript to improve clarity.2) Extensive analysis of the modeling.3) Clearer distinction from the prior work (including not copying prior methods) by the same group.

We have revised our manuscript extensively, based on the three points of essential revisions above and following the reviewers’ suggestions, as detailed in our point-by-point reply. Furthermore, we have added a new data set of neural recording from a larger social group (four bats) engaged in free social interactions, which confirmed the predictions of our model.

Please see the individual reviewer comments for guidance on how to rewrite the manuscript.As for Item #2, Reviewer #1 raised most of the concerns, and they were discussed among all of the reviewers who concur. During the discussion, it was raised that perhaps the findings of the model didn't really increase understanding past the basic data finding in Figure 1. In particular, there are a huge number of fitting parameters, such as the "b" values. The b values seem to effectively change the DC set point of the model for each hand-scored behavioral epoch, viaa* = - C^-}^_b_(this is not explained in the manuscript). This is roughly equivalent to taking all events ("grooming", "mating", etc), finding nominal DC point of the associated LFP, and subtracting it off, i.e. the b terms add an affine term that biases the linear dynamics to a nominal value. This makes sense, since power is a nonnegative quantity, and adding this DC offset effectively linearizes an affine system around a new operating point for each behavioral epoch, allowing for linear regression on the time constants.We could not determine how the authors fit the model parameters. If the parameters were optimized to reproduce the predictions, as opposed to fit the time domain data, for example, the model would be unconvincing and unfortunately not suitable for eLife; there are simply too many model DOFs and too few terms to predict, and so it is a foregone conclusion that you can fit the outcome.This speaks to the broader point that there was agreement among the reviewers that there was no model validation. For example, we could not find reported even a single time-domain trace comparing the model with data.Again, how does the model move past Figure 1 data analysis? Instead of fitting a complex, hierarchical model, would it be sufficient to analyze the constants by looking at the autocorrelation functions for each term (a2 + a1)/2 and (a2-a1)/2? The paper basically does this in terms of the PSD (Fourier transform of the autocorrelation functions).Reviewer #1:

We thank the reviewer for the helpful comments and constructive criticism, which has helped us better understand our model and improve our manuscript.

This paper provides experimental and modeling analysis of the inter-brain coupling of socially interacting bats, and reports that coordinated brain activity evolves at a slower time scale than the activity describing the differences. Specifically, the paper finds that there is an attracting submanifold corresponding to the mean (or "common mode") of neural activity, and that the dynamics in the orthogonal eigenmode, corresponding to the difference in brain activity, decays rapidly. These rapid decays in the difference mode are referred to as "catch up" activity.There are two main findings:1) Neural activity (especially higher frequency LFP activity in the 30-150Hz range) is modulated by social context. Specifically, the ratio of the averaged, moment-to-moment MEAN:DIFF ratio is much higher when the bats are in a single chamber, clearly indicating that the animals are coordinating their neural activity. This change also seems to hold -- although not as striking -- in lower-frequency LFP and spiking activity.2) The time scales of the mean vs. difference dynamics are segregated: the "difference dynamics" evolve at a faster time scale than "similarity dynamics", seems to be well supported.The basic finding is presented in Figure 1. The rest of the paper is focused on a modeling study to garner further insight into the dynamics.Weaknesses:This is an entirely phenomenological paper, and while it claims to garner "mechanistic insight", it is unclear what that means.

We regret not clarifying sufficiently what we meant by “mechanistic insight.” The insight is the following: functional across-brain coupling acts as positive feedback to the mean component of neural activity, which amplifies it and slows it down; at the same time, it acts as negative feedback to the difference component, which suppresses it and speeds it up. Thus, findings (1) and (2) in the reviewer’s summary above can be explained by the same model mechanism.

As the reviewer pointed out below, the details of the model are complex, which could have made the simple mechanism above opaque. Thus, we analyzed two simplified versions of the model to make the mechanistic insight clear. This is detailed below in our response to the reviewer’s comment on model complexity.

The basic idea of the model is simple and somewhat interesting, but the details are extremely complex. There are many examples of this, but the method used to "regress out" the behavior was very hard to interpret.

The method for regressing out behavior was described in Materials and methods, and we regret having neglected to reference it in the main text. We now reference it at the first instance in the main text where this is relevant.

On the face of it, the model is extremely simple: a two-state linear dynamical system. However, this simplistic description buries extreme complexity. The model is extremely complex as involves a large number of parameters (e.g., time switching 'b' values, the values of which are completely unclear), the switching over time of these parameters based on hand-scored animal behavioral state, and the complex mix of markovian and linear dynamical systems theoretic results.

As the reviewer pointed out, the core of the model is very simple: a linear dynamical system that models neural activity coupling. The model mechanism of positive and negative feedback, which is responsible for reproducing the two experimental results summarized by the reviewer above, is contained in this core (see Materials and methods for details). On top of this, the model has a layer of complexity, involving a Markov chain model of behavior and a large number of behavioral parameters. This layer of complexity is independent from the feedback mechanism of the core of the model. Thus, while it makes the model more biologically realistic, it is not required to reproduce the two main experimental results. To explicitly show this, and to better understand the dependence of model behavior on its parameters, we analyzed two reduced versions of the model.

The first reduced model replaces the behavioral inputs with white noise. The original model is τdadt=Ca+b, where a is neural activity, C=(−CSCICI−CS) is the coupling matrix, b is behavioral modulation, and τ is a time constant. b is where the complexity lies, as it is simulated using a Markov chain and involves many parameters. To strip away this layer of complexity, we replaced b with noise having a simple structure, namely, the mean and difference components of b having identical, flat power spectra. Importantly, this noise input does not induce correlation between bats, and it amounts to inputs of the same magnitude and same timescales to the mean and difference components of a.

The resulting reduced model has only two parameters, the functional self-coupling CS and functional across-brain coupling CI (for simplicity, τ can be absorbed into the other parameters). We are interested in the two results the reviewer summarized above: (1) the mean component of neural activity having a larger variance than the difference component; (2) the mean component having a slower timescale than the difference component. In the manuscript, these are respectively quantified using the variance ratio and the power spectral centroid ratio of the mean and difference components. The reduced model allowed us to derive analytical expressions for these two quantities (see Materials and methods for details). We found that they have very simple dependence on the functional coupling parameters: the variance ratio (mean variance divided by difference variance) is approximately CS+CICS−CI, and the centroid ratio (mean centroid divided by difference centroid) is approximately CS−CICS+CI.

This parameter dependence is visualized in Figure 3I-J (note that the color maps are in log scale, and the white spaces are regions where the model is unstable). In the experimental data, the mean component had larger variance and lower power spectral centroid than the difference component. This corresponds to the parameter regime of 0<CI<CS (enclosed by dashed lines in Figure 3I-J). Thus, a positive CI acts as positive feedback to the mean component and negative feedback to the difference component, modulating their variance and timescales in opposite directions. This is consistent with the analysis of the original model in Materials and methods. In the revised manuscript, we’ve now added analysis of this reduced model to the Results section.

The reviewer has stated a concern regarding the large number of parameters that set the input level according to behavioral state (bresting, bsocial grooming, bfighting, etc.). These parameters are important for ensuring that the model outputs realistic levels of behaviorally modulated neural activity (discussed below in our reply regarding model fit), but they are not important for the main results on variance and timescales. To demonstrate this, we studied a second reduced model. This model is identical to our original model except that, for each simulation, each of the behavior parameters (bfighting, etc.) was independently drawn from the uniform distribution from 0 to 1. Despite the completely random behavioral parameters, this reduced model reproduces the variance and timescales results just like the original model, as shown in Author response image 1 (compare with Figure 3E-F).

**Author response image 1. sa2fig1:** 

To summarize, the reduced models allowed us to identify the simple parameter dependence of the modeling results, and showed that the simple linear dynamical system at the core of the original model is sufficient to reproduce the two main experimental observations.

Indeed, a fundamental weakness of the model is that the Markov chain is taken as an "input" to the 2-state linear systems model, as if somehow the neural state does not affect the state transitions.

Yes, this is a limitation of our model. We’ve added a discussion of this limitation, as well as future directions for overcoming it, in the Discussion section.

The reason we did not model neural control of behavioral transitions is that it is under-constrained by existing data. While the brain obviously controls behaviors, not every part of the brain controls every behavior. Of the 11 behaviors observed in this study, we do not know which of them is controlled by the bat frontal cortex, and we do not know how they might be controlled (i.e., what specific spatiotemporal activity patterns affects behaviors in what ways). Without this knowledge, it’s unclear how to implement neural control of behavior in the model. This knowledge requires perturbation studies (lesion, inactivation, or activity manipulation) to establish casual relationships from neural activity to specific behaviors in the bat, which will be an important future direction.

On the other hand, as the reviewer stated, our model included behavioral modulation of neural activity. It is well known that in mammals, arousal and movement modulate neural activity globally across cortex (McGinley et al., 2015, Neuron). Thus, given that different behaviors in general involve different levels of arousal and movement, our model included behavior-dependent modulation of frontal cortical neural activity.

Finally, for the reviewer’s convenience, we also quote below the paragraph addressing this issue in the revised Discussion.

“Another limitation of our model is the “open-loop” nature of the relationship between behavior and neural activity. Specifically, we modeled neural activity as being modulated by behavior, but behavior was modeled using a Markov chain that is independent from the neural activity. In reality, neural activity and behavior form a closed-loop, with different social behaviors being controlled by the neural activity of specific neural populations in specific brain regions. Thus, an important future direction is to close the loop by incorporating neural control of social behaviors into models of the inter-brain relationship in bats. This will require future experimental studies to identify which frontal cortical regions and populations in bats are necessary or sufficient to control social behaviors, as well as the detailed causal relationship from neural activity to social behavior. Furthermore, as social interactions can occur at multiple timescales, it will be interesting to investigate how these are controlled by neural activity at different timescales, and how those timescales are shaped by functional across-brain coupling. In summary, such a closed-loop model will shed light on how inter-brain activity patterns and dynamic social interactions co-evolve and feedback onto each other.”

Further, the Markov assumption is not rigorously tested.

We have now tested the Markov assumption, using the following methods. We compared three models of bat behaviors: (1) the independent model, where the behavioral state at a given time point is independent from the state at other time points; (2) the 1st-order dependency model, where the behavioral state at a given time point depends on the state at the previous time point only; (3) the 2nd-order dependency model, where the behavioral state at a given time point depends on the states at the two previous time points. The Markov assumption corresponds to model (2), which is used as a part of the main model of the paper. Note that models with longer time-dependencies (≥3) were not tested because the number of parameters grows exponentially with model order and our dataset is not large enough to fit them.

To compare the three models, we split the behavioral data into a training set and a test set, fitted each model on the training set (Laplace smoothing was used to avoid assigning zero probability to unobserved events), and calculated the log-likelihood of the test set under each model. Figure 3—figure supplement 2A-B shows the cross-validated likelihoods for the behavioral data of one-chamber (A) and two-chambers (B) sessions, which were fitted separately; circles and error bars are means and standard deviations across 100 random splits of the data into training and test sets.

As Figure 3—figure supplement 2A-B shows, the 1st-order model had the highest likelihood on average. This does not necessarily prove that bat behavior obeys the Markov assumption (if we had a lot more data, we might be able to fit better 2nd-order and higher-order models). But this does mean that, given the amount of data we have, the best model that we can fit is the 1st-order Markov chain. Thus, this result supports our usage of the Markov chain in the main model of the paper.

In the revised manuscript, the analysis is described in Materials and methods.

No model selecting or other model validation appears to be done.

To evaluate model fit, we simulated our model using experimentally observed behaviors (rather than simulating behaviors using a Markov chain), and compared the simulated neural activity with the experimentally observed activity (see Materials and methods for detailed procedures). The comparison for an example experimental session is shown in Figure 3—figure supplement 1, where we’ve plotted the experimentally observed neural activity and behaviors for bat 1 (A) and bat 2 (B), along with the simulated neural activity. The correlation coefficient between data and model are indicated above each plot. These are representative examples, as the average correlation over all sessions and bats is 0.72 (standard deviation is 0.10).

In evaluating model fit, we realized that the model in the original manuscript produced outputs with a DC offset different from that of the data. Thus, in the revised manuscript, we added one more behavioral parameter (bconstant) that adjusts the DC offset, which is a parameter that reflects the effect of a baseline arousal level on neural activity (Materials and methods). Note that, since the only effect of this parameter is to adjust the DC offset of neural activity, it does not change any of the results in the paper.

In short, the model, while very interesting, is so complex that it is literally impossible to evaluate. The authors report literally no shortcomings of their model. They do not report parameter estimation methods. They do not report fitting errors or other model validation metrics. The only evaluation is whether it can produce certain outputs that are similar to biological data. While the latter is certainly important, all models are wrong, and it essential to have a model simple enough to understand, both in terms of how it works and how it fails.

The comments on the complexity of the model and on fitting errors have been addressed above. Regarding parameter estimation methods, they were described in Materials and methods, and we regret having neglected to directly reference it in the original manuscript. We now reference it in the legend of Figure 3A which is the first place to introduce the parameters. Briefly, the behavioral parameters (bresting, bfighting, etc.) were simply chosen to be the average neural activity during the respective behaviors from the data; the other parameters were chosen by hand to roughly match the levels of activity from the data, keeping within the parameter regime of 0<CI<CS identified from the analyses. As we showed above, these parameters provide a reasonable fit to the data.

The reason we chose the parameters heuristically in this way, rather than by minimizing some error objective, is the following. Our goal was to build a model that could qualitatively reproduce the experimental findings in a robust manner, that is, without fine-tuning of parameters. Thus, we analyzed the model to understand how model behaviors depend on the parameters, and to identify the parameter regime that reproduces the qualitative trends seen in the data (Figure 3I-J; Materials and methods). Guided by these analyses, we chose parameters heuristically without algorithmic fine-tuning.

Finally, following suggestions from reviewer 1 and reviewer 3, we have added discussions of shortcomings of the models (the last two paragraphs of the Discussion). With these discussions of model limitations, along with the presentation of simple insights into model mechanism from the reduced models above, we believe we have now presented a model that is “simple enough to understand, both in terms of how it works and how it fails.”

In general, while the basic finding is fairly interesting, and the experiments and their findings are highly relevant to the field, the modeling and its explication fall short.It is not that it is wrong or bad; however, it is not clear that such a complex model increases our understanding beyond the experimental findings in Figure 1, and if it does, there has to be a major caveat that the model itself is not carefully vetted.

Based on the reviewer’s comments on the model’s complexity, we have analyzed reduced versions of the model to understand its simple underlying mechanisms, as described above. This goes beyond the experimental findings in Figure 1, as it provides a computational mechanism that could give rise to those experimental findings. Moreover, based on the reviewer’s comments, we have more carefully vetted the model, by evaluating model fit and testing different behavioral models that assume or doesn’t assume the Markov property. Finally, we now discuss caveats of the model in the Discussion section, including the open-loop nature of the model as pointed out by the reviewer.